# Exponential Convergence Guarantees for Iterative Markovian Fitting

**Marta Gentiloni Silveri**
École polytechnique
Route de Saclay, 91120 Palaiseau, France
`marta.gentiloni-silveri@polytechnique.edu`

**Giovanni Conforti**
Università degli Studi di Padova
Via Trieste, 63, 35131 Padova, Italia
`giovanni.conforti@math.unipd.it`

**Alain Durmus**
École polytechnique
Route de Saclay, 91120 Palaiseau, France
`alain.durmus@polytechnique.edu`

## Abstract

The Schrödinger Bridge (SB) problem has become a fundamental tool in computational optimal transport and generative modeling. To address this problem, ideal methods such as Iterative Proportional Fitting and Iterative Markovian Fitting (IMF) have been proposed—alongside practical approximations like Diffusion Schrödinger Bridge and its Matching (DSBM) variant. While previous work have established asymptotic convergence guarantees for IMF, a quantitative, non-asymptotic understanding remains unknown. In this paper, we provide the first non-asymptotic exponential convergence guarantees for IMF under mild structural assumptions on the reference measure and marginal distributions, assuming a sufficiently large time horizon. Our results encompass two key regimes: one where the marginals are log-concave, and another where they are weakly log-concave. The analysis relies on new contraction results for the Markovian projection operator and paves the way to theoretical guarantees for DSBM.

## 1 Introduction

Generative models play a foundational role in modern machine learning, with widespread applications ranging from image synthesis [RPG⁺21] and natural language generation [BMR⁺20] to molecular structure design [XLH⁺22]. These models aim to learn the underlying probability distribution of a given dataset and can generate new, high-fidelity samples that resemble the original data. Among the various generative frameworks, diffusion and flow-based models have emerged as particularly powerful, thanks to both their empirical success and theoretical grounding. These models typically rely on stochastic differential equations (SDEs) to gradually transform a simple prior (often Gaussian noise) into samples from a complex target distribution, often through a learned score function [SE19, HJA20, SSDK⁺21]. However, despite their effectiveness, traditional score-based generative models (SGMs) often suffer from long sampling times, due to the need for finely discretized time grids and numerically stable SDE integration over long time horizons.

An increasingly popular and theoretically grounded alternative is to cast generative modeling as a Schrödinger Bridge (SB) problem. Originally introduced by Erwin Schrödinger in the 1930s [Sch32], the SB problem arises from statistical physics and seeks the most likely evolution of a cloud of independent particles (typically Brownian) that interpolates between two observed empirical distributions $\mu, \nu \in \mathbb{R}^d$ at prescribed initial and final times. Formally, the SB problem consists in finding a path

39th Conference on Neural Information Processing Systems (NeurIPS 2025).

measure that minimizes the Kullback-Leibler (KL) divergence with respect to a reference diffusion process (usually a stationary Brownian motion), subject to fixed marginal constraints. Although originally motivated by principles from statistical mechanics, the SB problem is now known to be equivalent to a regularized version of classical optimal transport (OT), where an entropic term is added to the transport cost [Nut]. In this formulation, the KL divergence acts as a soft transport penalty, leading to the interpretation of the SB as entropy-regularized OT in path space.

In addition to the path space (dynamical) formulation, it admits a static reformulation, where the optimization is performed over couplings between the given marginals. The optimal coupling $\pi^\star$, often referred to as the Schrödinger bridge, can be used to sample joint initial and terminal states. Conditional trajectories are then obtained by interpolating these endpoints with standard Brownian bridges. When the initial distribution is taken as the standard Gaussian $\gamma^d$, solving the SB problem identifies the most likely diffusion process — in terms of KL divergence from the reference — whose marginals are $\gamma^d$ and $\nu$. As a result, this formulation enables sample generation from $\nu$ in finite time, and provides a compelling alternative to traditional score-based generative approaches.

Several algorithms have been developed to compute or approximate solutions to the generalized SB problem, especially when the reference process is not necessarily Brownian motion. Among these, two families of algorithms stand out: the Iterative Proportional Fitting (IPF) algorithm (also known as the Sinkhorn algorithm) [Cut13, Nut], and the Iterative Markovian Fitting (IMF) algorithm (also known as Iterated Diffusion Bridge Mixture) [Pel23, SDBCD24]. IPF, on one hand, can be interpreted as an alternative minimization scheme where the marginals constraints are alternatively relaxed, which leads to a sequence of forward and backward diffusion processes with the target measures $\mu$ and $\nu$ as initial distributions. On the other hand, IMF defines a sequence of stochastic interpolations between $\mu$ and $\nu$ processes and their Markov projections, building on the Diffusion Flow Matching framework [Pel21, AVE22] and extending it recursively. In addition, practical and approximate implementations of IPF and IMF have been proposed. In particular, the Diffusion Schrödinger Bridge algorithm [DBTHD21a] along with its Matching version [SDBCD24, Pel23] or the Iterative Proportional Maximum Likelihood (IPML) algorithm [VTLL21a]

Although empirical performance of IMF is promising, its theoretical analysis remains limited. In particular, existing works only provide asymptotic convergence results [SDBCD24, Pel23], and no non-asymptotic rates or guarantees are currently known. In this paper, we fill this gap by providing the first non-asymptotic convergence guarantees for IMF. Under mild assumptions on the reference diffusion process and for sufficiently large time horizon $T$, we derive explicit convergence bounds in KL divergence. Our analysis distinguishes two important regimes: strongly log-concave marginals and weakly log-concave marginals — the latter being a more general class that includes multimodal and non-convex distributions [Con24]. A central technical contribution of our work is the derivation of a new contraction estimate for the Markovian projection operator (see Theorem 4), which plays a fundamental role in controlling the error propagation over iterations. Our main results — Theorem 5 and Theorem 6 — establish the first quantitative convergence rates for Schrödinger-bridge-based generative models.

**Outline of the Paper.** Section 2 introduces the necessary background on the Schrödinger Bridge problem and details the Iterative Markovian Fitting (IMF) algorithm. Section 3 presents our main theoretical results under both the strongly log-concave (Section 3.1) and weakly log-concave (Section 3.2) regimes. Section 3.3 discusses connections to prior work. Full technical details and supporting lemmas are deferred to the supplement.

**Notation.** For a metric space $(\mathsf{E}, d)$, denote by $\mathcal{P}(\mathsf{E})$ the set of probability measures on $\mathsf{E}$ and by $\mathcal{B}(\mathsf{E})$ its Borel $\sigma$-field. Given two probability measures $\mu, \nu \in \mathcal{P}(\mathsf{E})$, the relative entropy (or KL-divergence) of $\mu$ with respect to $\nu$ is defined by $\mathrm{KL}(\mu|\nu) := \int \log(\mathrm{d}\mu/\mathrm{d}\nu)\mathrm{d}\mu$ if $\mu$ is absolutely continuous with respect to $\nu$, and $\mathrm{KL}(\mu|\nu) := +\infty$ otherwise. Note that we also consider an extension of this definition to the case where $\nu$ is only a $\sigma$-finite measure on $\mathsf{E}$. Similarly, the Fisher information of $\mu$ with respect to $\nu$ is defined by $\mathscr{I}(\mu|\nu) := \int \|\nabla \log(\mathrm{d}\mu/\mathrm{d}\nu)\|^2 \mathrm{d}\mu$ if $\mu$ is absolutely continuous with respect to $\nu$, and $\mathscr{I}(\mu|\nu) := +\infty$ otherwise. When, $\mathsf{E} = \mathbb{R}^d$, denote by $\Pi(\mu, \nu)$ the set of couplings between $\mu$ and $\nu$, *i.e.*, $\xi \in \Pi(\mu, \nu)$ if and only if $\xi$ is a probability measure on $\mathbb{R}^d \times \mathbb{R}^d$ and $\xi(\mathsf{A} \times \mathbb{R}^d) = \mu(\mathsf{A})$ and $\xi(\mathbb{R}^d \times \mathsf{A}) = \nu(\mathsf{A})$ for all measurable $\mathsf{A} \subseteq \mathbb{R}^d$. If $\mu$ and $\nu$ have finite second moment, the $2-$Wasserstein distance is defined by

$\mathscr{W}_2^2(\mu, \nu) := \inf_{\xi \in \Pi(\mu, \nu)} \int \|x - y\|^2 \, \mathrm{d}\xi(x, y)$. Denote by $\gamma^d$ the density of the standard Gaussian distribution on $\mathbb{R}^d$. With abuse of notation, we identify the standard Gaussian measure with its density. For $d \in \mathbb{N}$, denote by $\mathrm{Leb}^d$ the Lebesgue measure on $\mathbb{R}^d$. Denote by $\mathcal{C}_T = \mathcal{C}([0, T])$ the set of continuous functions defined on the time-interval $[0, T]$. Equipped with the $\mathrm{L}^\infty$-norm, $\|\cdot\|_\infty$, refer to it as Wiener space. Given $\mathbb{P} \in \mathcal{P}(\mathcal{C}_T)$ and $s, t \in [0, T]$, denote by $\mathbb{P}_t \in \mathcal{P}(\mathbb{R}^d)$, $\mathbb{P}_{s,t} \in \mathcal{P}(\mathbb{R}^{2d})$, $\mathbb{P}_{|s,t} \in \mathcal{P}(\mathcal{C}_T)$ and $\mathbb{P}_{[s,t]} \in \mathcal{P}(\mathcal{C}([s, t]))$ respectively the marginal time distribution at time $t$, the joint law at times $s$ and $t$, the conditional distribution with respect to the marginals at times $s$ and $t$, and the the restriction to the time sub-interval $[s, t]$ of $\mathbb{P}$. Also, we denote by $\mathbb{P}^{\mathrm{R}} \in \mathcal{P}(\mathcal{C}_T)$ the reverse-time measure of $\mathbb{P}$, *i.e.*, the path-measure defined, for any $\mathsf{A} \in \mathcal{B}(\mathcal{C}_T)$, as $\mathbb{P}^{\mathrm{R}}(\mathsf{A}) = \mathbb{P}(\mathsf{A}^{\mathrm{R}})$ where $\mathsf{A}^{\mathrm{R}} := \{t \mapsto \omega(T - t) \; : \; \omega \in \mathsf{A}\}$. Given two matrices $\mathrm{A}, \mathrm{B} \in \mathbb{R}^{d \times d}$ we write $\mathrm{A} \succeq \mathrm{B}$ if $\mathrm{A} - \mathrm{B}$ is positive semi-definite. Given two vectors $x, y \in \mathbb{R}^d$, we denote by $\langle x, y \rangle$ and $\|x\|$ the standard scalar product between $x$ and $y$ and the Euclidean norm of $x$. Last, denote by $\mathrm{Lip}_{\leq 1}(\mathbb{R}^d)$ the set of Lipschitz functions $f : \mathbb{R}^d \to \mathbb{R}$ with Lipschitz constant smaller than 1.

## 2 Schrödinger Bridge Problem and Iterative Markovian Fitting

In this section, we present the SB problem in details and its resolution through IMF.

**Schrödinger Bridge problem.** The first key tool for defining the SB problem is a path measure $\mathrm{R}^U$ on $\mathcal{C}_T$, referred to as the reference measure. We consider in this paper for $\mathrm{R}^U$ the distribution of the process $(\mathbf{X}_t)_{t \in [0, T]}$ solution of the Langevin dynamics

$$\mathrm{d}\mathbf{X}_t = -\nabla U(\mathbf{X}_t)\mathrm{d}t + \sqrt{2}\mathrm{d}B_t \,, \quad t \in [0, T] \,, \quad \mathbf{X}_0 \sim \mathbf{m}(\mathrm{d}x) \propto \exp(-U(x))\mathrm{d}x \,, \quad (1)$$

where $U : \mathbb{R}^d \to \mathbb{R}$ is a potential. We also consider the special case $U \equiv 0$ which corresponds to taking as $\mathrm{R}^U$ the measure on $\mathcal{C}_T$ associated with the Brownian motion initialized at the volume measure [Léo13, Annexe A]. We make the following assumption on (1)

**H1.** *Either $U = 0$ or the Langevin system (1) has a unique stationary solution.*

Under mild assumptions on $U$, **H**1 holds; see e.g., [Ken78].

Equipped with $\mathrm{R}^U$, given the two marginal distributions $\mu, \nu \in \mathcal{P}(\mathbb{R}^d)$, the SB problem can be expressed as the following minimization problem:

$$\text{minimize } \mathrm{KL}(\mathbb{P}|\mathrm{R}^U) \,, \text{ under the constraint } \mathbb{P} \in \mathcal{P}(\mathcal{C}_T) \,, \; \mathbb{P}_0 = \mu \,, \; \mathbb{P}_T = \nu \,. \quad (2)$$

This formulation is known as the Dynamical Schrödinger Bridge Problem, where the optimization is performed over measures defined on the path space. Interestingly, this dynamic problem has an equivalent static formulation [Léo13, Proposition 2.3]:

$$\text{minimize } \mathrm{KL}(\pi|\mathrm{R}_{0,T}^U) \,, \text{ under the constraint } \pi \in \Pi(\mu, \nu) \,, \quad (3)$$

where $\mathrm{R}_{0,T}^U$ denotes the joint law between the marginal distributions at times $t = 0$ and $t = T$ of $\mathrm{R}^U$. This formulation is known as the Static Schrödinger Bridge problem. To ensure existence of a unique solution for (2)-(3), we consider the following assumption.

**H2.** *There exists (at least) a coupling $\pi \in \Pi(\mu, \nu)$ such that $\mathrm{KL}(\pi|\mathrm{R}_{0,T}^U) < +\infty$.*

Under **H**2, [Nut, Theorem 2.1.] shows that problem (3) admits a unique solution, called the *Schrödinger Bridge*, which can be expressed as

$$\pi^\star(\mathrm{d}x, \mathrm{d}y) = \exp\left(-\varphi(x) - \psi(y)\right) \mathrm{R}_{0,T}^U(\mathrm{d}x, \mathrm{d}y) \,, \quad (4)$$

where $\varphi, \psi : \mathbb{R}^d \to (-\infty, +\infty]$.

Consider the bridge $\mathrm{bR}^U$ associated with $\mathrm{R}^U$, *i.e.*, the Markov kernel on $\mathbb{R}^{2d} \times \mathcal{C}_T$ corresponding to the conditional distribution of $(\mathbf{X}_t)_{t \in [0, T]}$ given $(\mathbf{X}_0, \mathbf{X}_T)$ and therefore satisfying for any $\mathsf{A} \in \mathcal{B}(\mathcal{C}_T)$, $\mathrm{R}^U(\mathsf{A}) = \int \mathrm{d}\mathrm{R}_{0,T}^U(x_0, x_T)\mathrm{bR}^U((x_0, x_T), \mathsf{A})$. Equipped with $\mathrm{bR}^U$, we can easily construct an optimal solution for (2) based on (4). Indeed, denoting by $\mathbb{P}^\star$ the path distribution of $(X_t^\star)_{t \in [0, T]}$

such that $(X_0^\star, X_T^\star) \sim \pi^\star$ and $(X_t^\star)_{t \in [0,T]} \sim \mathrm{bR}^U((X_0^\star, X_T^\star), \cdot)$, and using that for any $\mathbb{P} \in \mathcal{P}(\mathcal{C}_T)$ it holds [Léo13, Annexe A]

$$\mathrm{KL}(\mathbb{P}|\mathrm{R}^U) = \mathrm{KL}(\mathbb{P}_{0,T}|\mathrm{R}_{0,T}^U) + \mathbb{E}[\mathrm{KL}(\mathbb{P}_{|0,T}(\cdot|(\bar{X}_0, \bar{X}_T))|\mathrm{R}_{|0,T}^U(\cdot|(\bar{X}_0, \bar{X}_T))))] \, ,$$

with $(\bar{X}_0, \bar{X}_T) \sim \mathbb{P}_{0,T}$, we immediately get that $\mathbb{P}^\star$ is a solution for (2). For further details on SB, we refer to the two surveys [Léo13] and [Nut].

One of the most famous scheme to solve (2) (or equivalently (3)) is the Iterative Proportional Fitting (IPF) algorithm, also known as the Sinkhorn algorithm. This scheme roughly consists in solving this problem, relaxing the constraints on one of the marginals alternatively. However, these steps are typically intractable but, leveraging learning and statistical techniques, [DBTHD21b, VTLL21a] propose practical implementation for IPF.

Besides IPF, a recent class of algorithms have recently emerged as alternative for solving (2). They can be interpreted as an extension of Diffusion Flow Matching models that we now present.

**Diffusion Flow Matching Models.** Given the two marginal distributions $\mu, \nu \in \mathcal{P}(\mathbb{R}^d)$, Diffusion Flow Matching (DFM) models learn to transform samples from $\mu$ into samples from $\nu$ by following a continuous stochastic flow. The key idea is to construct an interpolating process, or bridge, between $\mu$ and $\nu$. To formalize this, again, we consider the bridge $\mathrm{bR}^U$ associated to (1) and a coupling $\pi_{0,T} \in \Pi(\mu, \nu)$. We then define, the resulting stochastic interpolant $(Y_t)_{t \in [0,T]}$ as

$$(Y_0, Y_T) \sim \pi_{0,T}, \quad (Y_t)_{t \in [0,T]}|(Y_0, Y_T) \sim \mathrm{bR}^U((Y_0, Y_T), \cdot) \, . \tag{5}$$

The process $(Y_t)_{t \in [0,T]}$ provides a smooth, random evolution from $\mu$ to $\nu$, but it is not a Markov process in general and does not satisfy a simple Stochastic Differential Equation (SDE), making direct sampling difficult. However, we can consider the non-homogeneous Markov diffusion process $(X_t^{(1)})_{t \in [0,T)}$, solution of the SDE

$$\mathrm{d}X_t^{(1)} = f_t^{(1)}(X_t^{(1)})\mathrm{d}t + \sqrt{2}\mathrm{d}B_t \, , \quad t \in [0,T) \, , \quad X_0^{(1)} \sim \mu \, , \tag{6}$$

where, for any $t \in [0,T)$, the *mimicking drift* is given, for any $y_t \in \mathbb{R}^d$, by

$$f_t^{(1)}(y_t) = \mathbb{E}\left[\overrightarrow{\phi}_t(Y_t, Y_T)\big|Y_t = y_t\right] - \nabla U(y_t) \, . \tag{7}$$

Here, for any $t \in [0,T)$, the vector field $\overrightarrow{\phi}_t$ is defined, for any $y_t, y_T \in \mathbb{R}^d$, by

$$\overrightarrow{\phi}_t(y_t, y_T) = 2\nabla_{y_t} \log p_{T|t}^U(y_T|y_t) \, , \tag{8}$$

where $p_{T|t}^U$ denotes the conditional density with respect to the Lebesgue measure of $\mathbf{X}_T$ given $\mathbf{X}_t$. Indeed, under the technical conditions **T**1 and **T**2 given in the supplement, it can be shown (see Theorem 3 in Appendix B) that the Markov process $(X_t^{(1)})_{t \in [0,T]}$ preserves the same marginal distributions as $(Y_t)_{t \in [0,T)}$, *i.e.*,

$$X_t^{(1)} \overset{\mathrm{dist}}{=} Y_t \, , \quad t \in [0,T] \, . \tag{9}$$

In particular, $X_0^{(1)} \sim \mu$ and $X_T^{(1)} \sim \nu$, that is $(X_t^{(1)})_{t \in [0,T]}$ interpolates between $\mu$ and $\nu$. This process is known as the Markovian Projection of the stochastic interpolant and originates from [Gyö86] and [Kry84].

In its most conventional implementation, DFM models typically consider either $U \equiv 0$ or a quadratic potential $U$, corresponding to an Ornstein–Uhlenbeck process. In such cases, the conditional log-density $(y_t, y_T) \mapsto \log p_{T|t}^U(y_T|y_t)$ is available in closed form. The drift function $f_t^{(1)}$ is then the solution to a regression problem, which can be learned from samples of the process $(Y_t)_{t \in [0,T]}$, often using a neural network approximation. In practice, DFMs generate approximate samples from the target distribution $\nu$ by applying the Euler–Maruyama scheme to discretize the SDE in (6), substituting the true drift $f_t^{(1)}$ with its learned approximation. This approach offers an efficient and trainable method to transform samples from $\mu$ into samples from $\nu$ via a continuous stochastic flow.

It turns out that DFM models can be extended to solve the SB problem (2) as we now describe.

**Iterative Markovian Fitting**   Iterative Markovian Fitting (IMF) (also known as Iterated Diffusion Bridge Mixture) [SDBCD24, Pel23] is an algorithm designed to iteratively approximate the Schrödinger Bridge. It builds upon the Diffusion Flow Matching (DFM) framework and can be interpreted as a recursive extension of it. At a high level, a DFM model takes a given coupling $\pi_{0,T} \in \Pi(\mu, \nu)$ as initial input, constructs the corresponding stochastic interpolant $(Y_t)_{t \in [0,T]}$ (5), computes its Markovian projection $(X_t^{(1)})_{t \in [0,T]}$ (6), and outputs a path measure $\mathbb{P}^{(1)}$ and a new coupling $\pi_{0,T}^{(1)} = \mathbb{P}_{0,T}^{(1)} \in \Pi(\mu, \nu)$ corresponding to the distribution of $(X_t^{(1)})_{t \in [0,T]}$ and the pair $(X_0^{(1)}, X_T^{(1)})$ respectively. By iterating this procedure multiple times, we obtain the IMF algorithm whose pseudo-code is provided in Algorithm 1.

---

**Algorithm 1:** Iterative Markovian Fitting

---

**Input:** Initial coupling $\pi_{0,T}^{(0)} \in \Pi(\mu, \nu)$ and Langevin bridge $\mathrm{bR}^U$ associated with (1)
**For each iteration** $k = 0, \ldots, N-1$:
  **Step 1.  Stochastic Interpolant Update.**
      Define $(Y_t^{(k+1)})_{t \in [0,T]}$ as:

$$\left(Y_0^{(k+1)}, Y_T^{(k+1)}\right) \sim \pi_{0,T}^{(k)}, \quad Y_{[0,T]}^{(k+1)} \big| \left(Y_0^{(k+1)}, Y_T^{(k+1)}\right) \sim \mathrm{bR}^U\left(\left(Y_0^{(k+1)}, Y_T^{(k+1)}\right), \cdot\right).$$

  **Step 2.  Markovian Projection Update.**

    • Consider $\mathbb{P}^{(k+1)}$ as the path-distribution of the solution $(X_t^{(k+1)})_{t \in [0,T]}$ to the SDE:

$$\mathrm{d}X_t^{(k+1)} = f_t^{(k+1)}(X_t^{(k+1)})\mathrm{d}t + \sqrt{2}\mathrm{d}B_t, \quad X_0^{(k+1)} \sim \mu, \tag{10}$$

     where:

$$f_t^{(k+1)}(y_t) = \mathbb{E}\left[\overrightarrow{\phi}_t\left(Y_t^{(k+1)}, Y_T^{(k+1)}\right)\big| Y_t^{(k)} = y_t\right] - \nabla U(y_t), \tag{11}$$

     and $\overrightarrow{\phi}_t$ is given in (8).
    • Set $\pi_{0,T}^{(k+1)} = \mathbb{P}_{0,T}^{(k+1)}$.
**End For**
**Output:** The path-distribution $\mathbb{P}^{(N)}$ and the coupling $\pi_{0,T}^{(N)} \in \Pi(\mu, \nu)$

---

It has been shown [Léo13, LRZ14] that the SB coupling $\pi^\star$, defined in (4), is the unique fixed point from the two Steps defining Algorithm 1, which can therefore be interpreted as a fixed point algorithm. Therefore, the sequences $\{\mathbb{P}^{(n)}\}_{n \in \mathbb{N}}$ and $\{\pi_{0,T}^{(n)}\}_{n \in \mathbb{N}}$ defined in Algorithm 1 are expected to converge to the SB solutions $\mathbb{P}^\star$ and $\pi^\star$. In particular, asymptotic convergence has been established in [SDBCD24, Pel23] under appropriate conditions. Further discussions on IMF, and its connection with Sinkhorn and Diffusion Schrödinger Bridge, are postponed in Appendix C.

During computational implementation, at each step $k$, the mimicking drift $f_t^{(k+1)}$ defined in (11) is learned by solving a regression problem using neural networks, following the same approach as in DFMs. This leads to the first version of the Diffusion Schrödinger Bridge Matching (DSBM) algorithm. However, from a practical standpoint, it has been observed that to avoid bias accumulation, it is beneficial to leverage not only the fact that $\mathbb{P}^{(k+1)}$ is the law of a diffusion process but also its time-reversal. Denote by $\tilde{\mathbb{P}}^{(k+1)}$ the distribution of the time reversal $(\tilde{X}_t^{(k+1)})_{t \in [0,T]}$ of $(X_t^{(k+1)})_{t \in [0,T]}$ defined for any $t \in [0,T]$ by $\tilde{X}_t^{(k+1)} = X_{T-t}^{(k+1)}$. It can be shown that $(\tilde{X}_t^{(k+1)})_{t \in [0,T)}$ is solution of the SDE (see Proposition 1 in Appendix D)

$$\mathrm{d}\tilde{X}_t^{(k+1)} = g_t^{(k+1)}(\tilde{X}_t^{(k+1)})\mathrm{d}t + \sqrt{2}\mathrm{d}B_t, \quad t \in [0,T), \quad \tilde{X}_0^{(k+1)} \sim \nu, \tag{12}$$

where, for any $t \in [0,T)$, the drift function is given, for any $y_{T-t} \in \mathbb{R}^d$, by

$$g_t^{(k+1)}(y_{T-t}) = \mathbb{E}\left[\overleftarrow{\phi}_t\left(Y_0^{(k+1)}, Y_{T-t}^{(k+1)}\right)\big| Y_{T-t}^{(k)} = y_{T-t}\right] + \nabla U(y_{T-t}), \tag{13}$$

and, for any $t \in [0, T)$, the vector field $\overleftarrow{\phi}_t$ is defined, for any $y_0, y_{T-t} \in \mathbb{R}^d$, by

$$\overleftarrow{\phi}_t(y_0, y_{T-t}) = 2\nabla_{y_{T-t}} \log p^U_{T-t|0}(y_{T-t}|y_0) \,.$$

Since up to time inversion, $\tilde{\mathbb{P}}^{(k+1)}$ and $\mathbb{P}^{(k+1)}$ are equivalent, an alternative to Step 2 in Algorithm 1 is to consider $\tilde{\mathbb{P}}^{(k+1)}$ instead of $\mathbb{P}^{(k+1)}$, and the resulting coupling $(\tilde{X}^{(k+1)}_T, \tilde{X}^{(k+1)}_0) \in \Pi(\mu, \nu)$. While this substitution is equivalent for the idealized IMF scheme, it becomes non-trivial when approximating the mimicking drifts. Indeed, comparing (10)–(11) with (12)–(13), we observe that the two processes are initialized at different distributions: the forward diffusion starts from $\mu$, while the time-reversed process starts from $\nu$. Alternating between Step 2 and its time-reversed variant thus mitigates the bias introduced by using approximate rather than exact mimicking drifts. This strategy leads to the final version of the DSBM algorithm proposed in [SDBCD24, Pel23]. Further details are provided in Appendix D.

## 3 Main Results

In this section, we establish exponential convergence guarantees in Kullback–Leibler (KL) divergence for the IMF Algorithm 1, under two different settings: the strongly log-concave and the weakly log-concave regimes. Although [SDBCD24, Theorem 8] and [Pel23, Theorem 2] demonstrate that, under strong assumptions, the IMF sequence $\{\pi^{(n)}_{0,T}\}_n$ converges asymptotically in KL divergence to the Schrödinger Bridge, i.e., $\lim_{n \to +\infty} \mathrm{KL}(\pi^{(n)}_{0,T} | \pi^\star) = 0$, neither work provides quantitative bounds on the convergence rate. This section aims to fill that gap.

Before proceeding further, we introduce an assumption on the reference measure (1), which will be considered in both the strong and weak log-concavity settings. Recall that for $0 < s < t$, we denote by $p^U_{t|s}$ the conditional distribution of $\mathbf{X}_t$ given $\mathbf{X}_s$ where $(\mathbf{X}_t)_{t \geq 0}$ is the unique stationary solution of (1) under **H1**.

**H3.** *There exists $L_U > 0$ such that, for any $0 \leq s < t \leq T$ and any $x, y \in \mathbb{R}^d$, the maps $z \mapsto \nabla_y \log p^U_{t|s}(z|y)$ and $z \mapsto \nabla_y \log p^U_{t|s}(y|z)$ are $L_U(s,t)$-Lipschitz with*

$$L_U(s,t) \leq \frac{L_U}{2(t-s)} \,.$$

**Remark 1.** *First note in the case where for any $0 \leq s < t \leq T$, $(z,x) \mapsto \log p^U_{t|s}(y|z)$ is $\mathrm{C}^2$, then* **H3** *is equivalent to the fact that $(z,y) \mapsto \nabla_z \nabla_y \log p^U_{t|s}(y|z)$ is bounded by $L(s,t)$.*

**Remark 2.** *We observe that in the common setting where $U \equiv 0$, the above condition is satisfied. Indeed, in that case, it holds*

$$p^U_{t|s}(y|x) = \frac{1}{(4\pi(t-s))^{d/2}} \exp\left(-\frac{\|y-x\|^2}{4(t-s)}\right) \,, \quad t,s \in [0,T] \,, \quad s < t \,, \quad x,y \in \mathbb{R}^d \,. \quad (14)$$

*Therefore, for any $0 \leq s < t \leq T$ and any $x, y \in \mathbb{R}^d$, they hold*

$$\nabla_x \log p^U_{t|s}(z|x) = \frac{z-x}{2(t-s)} \,, \quad \nabla_y \log p^U_{t|s}(y|z) = -\frac{y-z}{2(t-s)} \,.$$

*Similar conclusions hold for Ornstein-Ulhenbeck processes, i.e., as $U(x) = \langle \mathbf{A}x, x\rangle$. Indeed, if this is the case, then $\nabla U(x) = 2\mathbf{A}x$ and the SDE (1) becomes a linear Ornstein–Uhlenbeck (OU) process:*

$$d\mathbf{X}_t = -2\mathbf{A}\mathbf{X}_t dt + \sqrt{2}dB_t \,, \quad t \in [0,T] \,, \quad \mathbf{X}_0 \sim \mathbf{m}(dx) \,.$$

*This process admits a Gaussian transition density. More specifically,*

$$p^U_{t|s}(z|x) = \frac{1}{\sqrt{(2\pi)^d \det \Sigma_{t|s}}} \exp\left(-\frac{1}{2}(z - m_{t|s}(x))^\top \Sigma^{-1}_{t|s}(z - m_{t|s}(x))\right),$$

*where*

$$m_{t|s}(x) = \mathrm{e}^{-2\mathbf{A}(t-s)}x \,, \quad \Sigma_{t|s} = 2\int_s^t e^{-2\mathbf{A}(t-u)}\mathrm{e}^{-2\mathbf{A}^\top(t-u)}du \,.$$

*Therefore, it holds*

$$\nabla_x \log p_{t|s}^U(z|x) = \left(\nabla_x m_{t|s}(x)\right)^\top \Sigma_{t|s}^{-1}(z - m_{t|s}(x)) = \mathrm{e}^{-2\mathbf{A}(t-s)\top}\Sigma_{t|s}^{-1}(z - \mathrm{e}^{-2\mathbf{A}(t-s)}x) \ .$$

*It follows that the map $z \mapsto \nabla_x \log p_{t|s}^U(z|x)$ is affine, hence Lipschitz. Being*

$$\|\nabla_z \nabla_x \log p_{t|s}^U(z|x)\| \leqslant \|\mathrm{e}^{-2\mathbf{A}(t-s)}\| \cdot \|\Sigma_{t|s}^{-1}\| \ ,$$

*with*

$$\Sigma_{t|s}^{-1} \preceq \frac{1}{2(t-s)}\left(\mathrm{e}^{-2\mathbf{A}(t-s)}\mathrm{e}^{-2\mathbf{A}^\top(t-s)}\right)^{-1} \mathrm{Id} \ ,$$

*then, it holds*

$$\|\nabla_z \nabla_x \log p_{t|s}^U(z|x)\| \leqslant \|\mathrm{e}^{-2\mathbf{A}(t-s)}\| \cdot \|\Sigma_{t|s}^{-1}\| \leqslant \frac{L_U}{2(t-s)} \ ,$$

*with $L_U$ depending only on $\mathbf{A}$. This and Remark 1 complete the proof. We expect that Assumption **H 3** holds true also if $U$ is infinitely differentiable with bounded derivatives, but this problem is out of the scope of the paper and left for future work.*

In the sequel, for any measure $\zeta$ on $(\mathbb{R}^d, \mathcal{B}(\mathbb{R}^d))$ absolutely continuous with respect to the Lebesgue measure, we denote by $\mathfrak{p}_\zeta$ its density. Furthermore, we denote by $\mathbf{m}_{0,t,T}$ and $p_{0,t,T}^U$ the probability distribution and the density with respect to the Lebesgue measure respectively of $(\mathbf{X}_0, \mathbf{X}_t, \mathbf{X}_T)$ with $(\mathbf{X}_t)_{t \in [0,T]}$ as in (1), for $t \in (0,T)$.

### 3.1 Strongly Log-Concave Setting

We consider in this section the following additional assumptions.

**H4.** *The two distributions $\mu$ and $\nu$ are absolutely continuous with respect to the Lebesgue measure and have positive density functions $\mathfrak{p}_\mu$ and $\mathfrak{p}_\nu$. In addition, $x \mapsto \log \mathfrak{p}_\mu(x)$ and $x \mapsto \log \mathfrak{p}_\nu(x)$ are twice continuously differentiable. In addition, $(x_0, x_T) \mapsto -\log p_{0,t,T}^U(x_0, x_t, x_T)$ is twice continuously differentiable.*

**H5.** *(i) The two distributions $\mu$ and $\nu$ are strongly log-concave, i.e., there exist $\alpha_\mu, \alpha_\nu \in (0, +\infty)$ such that for any $x \in \mathbb{R}^d$,*

$$\nabla^2(-\log \mathfrak{p}_\mu)(x) \succeq \alpha_\mu \, \mathrm{Id} \ , \quad \nabla^2(-\log \mathfrak{p}_\nu)(x) \succeq \alpha_\nu \, \mathrm{Id} \ .$$

*(ii) There exists $\alpha \in [0, +\infty)$ such that for any $x_0, x_t, x_T \in \mathbb{R}^d$, $t \in (0,T)$,*

$$\nabla_{x_0, x_T}^2(-\log p_{0,t,T}^U)(x_0, x_t, x_T) \succeq \alpha \, \mathrm{Id} \ ,$$

*where $\nabla_{x_0, x_T}^2(-\log p_{0,t,T}^U)(x_0, x_t, x_T)$ denotes the Hessian with respect to the variables $(x_0, x_T)$.*

**Remark 3.** *We remark that in the common setting $U \equiv 0$ in (1), the part of **H4** and **H5** related to $-\log p_{0,t,T}^U$ holds. Indeed, when $U \equiv 0$, it follows from (14) that*

$$p_{0,t,T}^U(x_0, x, x_T) = \exp\left(-\frac{\|x_0 - x_T\|^2}{4T} - \frac{\|Tx - (T-t)x_0 - tx_T\|^2}{4tT(T-t)}\right) \ ,$$

*which makes it clear that $-\log p_{0,t,T}^U$ satisfies **H4** and **H5** since for any $x_0, x_t, x_T \in \mathbb{R}^d$ and $t \in (0,T)$, $\nabla_{x_0, x_T}^2[-\log p_{0,t,T}^U](x_0, x_t, x_T) \succeq 0$. The same conditions hold true when $U$ is quadratic, i.e., when $U(x) = \langle \mathbf{A}x, x \rangle$ for a positive definite matrix $\mathbf{A}$. In this setting, the process $(\mathbf{X}_t)_{t \in [0,T]}$ defined in (1) is a multivariate Ornstein–Uhlenbeck process. Consequently, for any $0 < t < T$, the joint distribution of the triple $(\mathbf{X}_0, \mathbf{X}_t, \mathbf{X}_T) \in \mathbb{R}^{3d}$ is a multivariate Gaussian, $(\mathbf{X}_0, \mathbf{X}_t, \mathbf{X}_T) \sim \mathrm{N}(\mu, \Sigma)$, for some mean vector $\mu \in \mathbb{R}^{3d}$ and for a positive definite covariance matrix $\Sigma \in \mathbb{R}^{3d \times 3d}$. This immediately implies that **H4** and **H5** are satisfied. Indeed, let $[\Sigma^{-1}]_{x_0, x_T}$ denote the principal submatrix of $\Sigma^{-1}$ corresponding to the variables $x_0$ and $x_T$. Then,*

$$\nabla_{x_0, x_T}^2\left(-\log p_{0,t,T}^U(x_0, x_t, x_T)\right) = [\Sigma^{-1}]_{x_0, x_T} \succeq \alpha \, \mathrm{Id} \ ,$$

*where the final inequality follows from the positive definiteness of $\Sigma$, which implies that all its principal submatrices–and thus those of $\Sigma^{-1}$–are also positive definite. Therefore, there exists $\alpha > 0$ such that the above inequality holds.*

This setting ensures robust stability properties, enabling strong convergence guarantees for IMF. The following theorem formalizes our result in this regime.

**Theorem 1.** *Assume **H**1 to **H**5. Let $\{\pi_{0,T}^{(n)}\}_{n\geq 1}$ be the IMF sequence defined in Algorithm 1. If $T > \max\{\alpha_\mu^{-1}, \alpha_\nu^{-1}\}$, then, for any $n \in \mathbb{N}$, it holds*

$$\mathrm{KL}(\pi_{0,T}^{(n)}|\pi^\star) \leq \left(\frac{L_U}{T(\alpha_\varphi + \alpha_\psi + \alpha)}\right)^n \mathrm{KL}(\pi_{0,T}|\pi^\star) ,$$

*where $\alpha_\varphi = \alpha_\mu - T^{-1}$ and $\alpha_\psi = \alpha_\nu - T^{-1}$.*

**Remark 4.** *Theorem 1 establishes that IMF converges to the Schrödinger Bridge $\pi^\star$ at an exponential rate in KL-divergence. The convergence rate is explicitly given by the factor*

$$\frac{L_U}{T(\alpha_\mu + \alpha_\nu + \alpha - 2T^{-1})} < 1 ,$$

*provided that $T > \max\{\alpha_\mu^{-1}, \alpha_\nu^{-1}, L_U(\alpha_\varphi + \alpha_\psi + \alpha)^{-1} + 2\}$. This shows that larger values of the convexity parameters $\alpha_\mu$ and $\alpha_\nu$, as well as a stronger convexity of the reference (via larger $\alpha$), lead to faster convergence. Conversely, as $T$ approaches $\max\{\alpha_\mu^{-1}, \alpha_\nu^{-1}, L_U(\alpha_\varphi + \alpha_\psi + \alpha)^{-1} + 2\}$ from above, the rate $\rho$ approaches 1, and the convergence becomes arbitrarily slow. This highlights the trade-off between the time horizon $T$ and the strength of log-concavity in determining the practical efficiency of the IMF algorithm.*

**Remark 5.** *We remark that the convergence bound provided in Theorem 1 is dimension-free in the same sense as in the literature on Langevin Monte Carlo: the bound does not depend explicitly on the ambient dimension, but rather on parameters of the marginal distributions, specifically, their log-concavity constants. While these parameters are independent of the dimension in some specific settings, they may degrade with dimension in more general scenarios.*

**Remark 6.** *We highlight that Theorem 1 can equivalently be reformulated in terms of the reference process's variance rather than time horizon, as done in [GSK+24]. This alternative presentation is a direct consequence of the Brownian scaling property ($B_t \sim \sigma B_{t/\sigma^2}$) and does not alter the asymptotic scaling behavior of the convergence rate.*

## 3.2 Weakly Log-Concave Setting

To extend our results beyond the strongly log-concave case, we consider a weaker notion of convexity, which is particularly relevant for machine learning applications involving multimodal distributions or heavy-tailed priors.

For a given differentiable vector field $\beta : \mathbb{R}^d \to \mathbb{R}^d$, we define its weak convexity profile as

$$\kappa_\beta(r) = \inf \left\{ \frac{\langle \beta(x) - \beta(y), x - y \rangle}{\|x - y\|^2} \; : \; \|x - y\| = r \right\} , \quad r > 0 .$$

This function quantifies non-uniform convexity lower bounds, capturing long-range dependencies in high-dimensional distributions. This definition is widely used in the study of Fokker–Planck equations via coupling methods (see [Ebe16]).
Define for any $L \geqslant 0$ and $r > 0$, $\vartheta_L(r) = 2\sqrt{L}\tanh(r\sqrt{L}/2)$.
In addition, for a distribution $\zeta$ absolutely continuous with respect to the Lebesgue measure with positive and continuously differential density function $\mathfrak{p}_\zeta$, we write $\kappa_\zeta := \kappa_{-\nabla \log \mathfrak{p}_\zeta}$.

**H6.** *The two distributions $\mu$ and $\nu$ are absolutely continuous with respect to the Lebesgue measure and have positive density functions $\mathfrak{p}_\mu$ and $\mathfrak{p}_\nu$. In addition, $x \mapsto \log \mathfrak{p}_\mu(x)$ and $x \mapsto \log \mathfrak{p}_\nu(x)$ are continuously differentiable. In addition, $(x_0, x_T) \mapsto -\log p_{0,t,T}^U(x_0, x_t, x_T)$ is continuously differentiable.*

**H7.** *(i) The two distributions $\mu$ and $\nu$ are weakly log-concave, i.e., there exist $\alpha_\mu, \alpha_\nu \in (0, +\infty), L_\mu, L_\nu > 0$ such that for any $r > 0$,*

$$\kappa_\mu(r) \geq \alpha_\mu - r^{-1}\vartheta_{L_\mu}(r) , \quad \kappa_\nu(r) \geq \alpha_\nu - r^{-1}\vartheta_{L_\nu}(r) .$$

*(ii) There exists $\alpha \in [0, +\infty), L \geq 0$ such that for any $t \in (0, T)$ and $x_t \in \mathbb{R}^d$, denoting by $\rho_{t,x_t} : (x_0, x_T) \mapsto -\nabla_{x_0,x_T} \log p_{0,t,T}^U(x_0, x_t, x_T)$ and $\kappa_{0,t,T} := \kappa_{\rho_{t,x_t}}$, it holds for any $r > 0$*

$$\kappa_{0,t,T}(r) \geq \alpha - r^{-1}\vartheta_L(r) .$$

**Remark 7.** *Note that **H**5 (ii) implies **H**7 (ii). Therefore, in light of Remark 3, we note that either in the classical and in the quadratic case **H**7 (ii) holds true.*

**Remark 8.** *Weakly log-concave distributions constitute a broad and expressive class of probability measures that can be seen as perturbations of log-concave distributions. For instance, this includes cases where the negative log-density, $-\log p$, has the form of a double-well potential*

$$-\log p(x) = \|x\|^4 - M\|x\|^2 + C \ ,$$

*for some $M > 0, C \in \mathbb{R}$. More generally, if $-\log p = V + W$ with $V$ strongly convex and $W$ Lipschitz with Lipschitz derivative, then $p$ is weakly log-concave. The class of weakly log-concave distributions also includes Gaussian Mixtures, as shown in [GSO25, Appendix A]. In fact, we remark that weak log-concavity property is also referred to as strong convexity at infinity in the literature related to the convergence of the Langevin diffusion. Indeed, such a hypothesis on the potential of the target distribution has been considered in various works including [Ebe16, CCAY$^+$18, BRE21, CZS22, MCJ$^+$19, CM25].*

By leveraging weak convexity properties, we derive exponential convergence results for IMF also in this more general setting.

**Theorem 2.** *Assume **H**1 to **H**4 and **H**6 and 7. Let $\{\pi_{0,T}^{(n)}\}_{n \geq 1}$ be the IMF sequence defined in Algorithm 1. If $T > \max\{\alpha_\mu^{-1}, \alpha_\nu^{-1}\}$, then, for any $n \in \mathbb{N}$, it holds*

$$\mathrm{KL}(\pi_{0,T}^{(n)}|\pi^\star) \leq \left(\frac{L_U}{T\mathrm{C}_{\varphi,\psi,U}}\right)^n \mathrm{KL}(\pi_{0,T}|\pi^\star) \ ,$$

*where*

$$\mathrm{C}_{\varphi,\psi,U} = 4(\alpha_\varphi + \alpha_\psi + \alpha) \left/ \exp\left(\frac{9\max\{L_\mu, L_\nu, L\}}{\alpha_\varphi + \alpha_\psi + \alpha}\right) \right. \ ,$$

*and $\alpha_\varphi = \alpha_\mu - T^{-1}, \alpha_\psi = \alpha_\nu - T^{-1}$.*

**Remark 9.** *Theorem 2 extends the exponential convergence guarantee of Theorem 1 to the setting where $\mu, \nu$, and the reference bridge are only weakly log-concave. The rate*

$$\frac{L_U}{4T(\alpha_\mu + \alpha_\nu + \alpha - 2T^{-1})} \exp\left(\frac{9\max\{L_\mu, L_\nu, L\}}{\alpha_\mu + \alpha_\nu + \alpha - 2T^{-1}}\right) < 1 \ ,$$

*provided that*

$$T > \max\left\{\alpha_\mu^{-1}, \alpha_\nu^{-1}, \frac{4L_U}{\alpha_\mu + \alpha_\nu + \alpha} \exp\left(\frac{18\max\{L_\mu, L_\nu, L\}}{\alpha_\mu + \alpha_\nu + \alpha}\right)\right\} \ ,$$

*remains exponential, but is dampened by the factor $\mathrm{C}_{\varphi,\psi,U}$, which worsens as the deviations from strong convexity (quantified by $L_\mu, L_\nu, L$) increase. This highlights that IMF still converges under weaker assumptions, but the speed is sensitive to the regularity and smoothness of the input and reference measures.*

**Remark 10.** *Theorem 2, as the previous one, can also be reformulated in terms of the reference process's variance rather than time horizon. The reasons are the same as those discussed in Remark 6. Moreover, the same considerations regarding the dependence on the dimension of the space, discussed in Remark 5 for the previous theorem, also apply here.*

### 3.3 Related Literature and Our Contribution

The study of Schrödinger Bridges and their applications has seen a surge of interest in recent years. Among the classical methods for solving the SB problem, a prominent role is played by the Sinkhorn algorithm, also known as the Iterative Proportional Fitting (IPF) procedure [For40b, Kul68a, Cut13]. More recently, advances in generative modeling have extended the reach of these techniques to continuous-time settings [Rü95, DBTHD21a]. Notably, the Diffusion Schrödinger Bridge (DSB) algorithm [DBTHD21b] and the Iterative Proportional Maximum Likelihood (IPML) method [VTLL21b] represent two pioneering efforts to approximate Schrödinger Bridge solutions via iterative procedures grounded in continuous-time IPF and time-reversal arguments. While

IPML leverages Gaussian processes and maximum likelihood estimation, DSB relies on score-matching techniques and can be viewed as a recursive extension of Score-Based Generative Models (SGMs) [SE19, HJA20, SSDK$^+$21]. SGMs focus on learning the score function of a stochastic differential equation (SDE), which is then used to simulate the time-reversed diffusion process for sample generation. In recent years, Flow Matching methods [Pel21, LAVE$^+$22, AVE23b, AVE23a, SCD24] have become an alternative to SGMs. These methods construct transport maps using ordinary or stochastic differential equations, bypassing the need for a forward noising process that converges to a reference distribution, as required in SGMs [SSDK$^+$21]. Unlike SGMs, Flow Matching methods interpolate between arbitrary probability distributions in finite time, offering greater flexibility in the transport mechanism and improved numerical stability. Building on the principles of Flow Matching, two recent iterative schemes have been proposed for approximating the Schrödinger Bridge: the Iterative Markovian Fitting (IMF) algorithm, also referred to as the Iterated Diffusion Bridge Mixture, and the Diffusion Schrödinger Bridge Matching (DSBM) algorithm [SDBCD23, Pel23]. In contrast to classical IPF-based methods [DBTHD21b], IMF and DSBM incorporate Markovian projections that preserve either Markovian and marginals' consistency across iterations. This refinement leads to improved empirical performance in generative modeling tasks and provides a more principled approach to entropy-regularized optimal transport. A key distinction between the two algorithms lies in how they handle estimation errors in the drift terms. While IMF does not account for such errors, DSBM explicitly corrects for them via neural networks and $L^2$-estimates and mitigates the resulting bias on the terminal marginal $\nu$ by alternating between forward and backward Markovian projections. This bidirectional strategy enhances the robustness of the generated samples and leads to more reliable approximations of the underlying SB solution. Although prior work has extensively analyzed the theoretical properties of score-based generative models [CDS25, GSO25], IPF-type Schrödinger Bridge approximations [CDG23, CCGT24], and flow matching techniques [SCD24, BDD23, AVE22], the Iterative Markovian Fitting (IMF) algorithm has so far lacked rigorous, non-asymptotic guarantees. Existing analyses provide only asymptotic convergence results [SDBCD24, Pel23], leaving open the question of its quantitative behavior in finite iterations. In this work, we address this gap by establishing the first non-asymptotic exponential convergence guarantees for the IMF algorithm. Under mild regularity assumptions on the reference process $R^U$ and the marginals $\mu$ and $\nu$, and in the regime of large time horizon $T$, we derive explicit convergence rates in terms of KL divergence. These results provide a theoretical foundation for the use of IMF in practical generative modeling applications.

## 4    Conclusion

We provided the first quantitative theoretical guarantees for the Iterative Markovian Fitting algorithm, under mild assumptions on the marginals and the reference measure. Our analysis covers both strongly log-concave and weakly log-concave distributions—the latter being a broad class that includes, for example, double-well potentials—and yields explicit convergence rates, the first in the literature for this setting. A key technical ingredient is the proof of novel contraction properties of Markovian projections, which are not only essential to our convergence results but also of independent mathematical interest. Our results hold in the regime where the time horizon $T$ is sufficiently large, while in the Schrödinger Bridge problem $T$ is typically fixed. Moreover, our analysis is purely theoretical and does not account for the mimicking drift estimation error or the discretization error arising in practical implementations. These limitations point to natural directions for future work, particularly the extension to finite-time settings and the inclusion of approximation errors in the theoretical analysis.

**Acknowledgments and Disclosure of Funding**

The work of Marta Gentiloni-Silveri has been supported by the Paris Ile-de-France Région in the framework of DIM AI4IDF. The work by Alain Durmus is partially funded by the European Union (ERC-2022-SYG-OCEAN-101071601). Views and opinions expressed are however those of the author(s) only and do not necessarily reflect those of the European Union or the European Research Council Executive Agency. Neither the European Union nor the granting authority can be held responsible for them.

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

# A   Technical Appendices and Supplementary Material

## Additional Notation

Denote by $\Pi \in \mathcal{P}(\mathcal{C}_T)$ the law on the Wiener space of the stochastic interpolant $(Y_t)_{t \in [0,T]}$ (5). For a given $t \in [0, T]$ and a given $x \in \mathbb{R}^d$, denote by $\Pi_t$ the marginal time distribution at time $t \in (0, T)$ of $Y_t$, *i.e.*,

$$\Pi_t := \mathcal{L}(Y_t) \in \mathcal{P}(\mathbb{R}^d) \ .$$

It follows from the very definition (5) of stochastic interpolant that $\Pi_t$ is absolutely continuous with respect to the Lebesgue measure with density given, for any $x_t \in \mathbb{R}^d$, by

$$p_t^Y(y_t) = \int_{\mathbb{R}^d \times \mathbb{R}^d} p_{t|(0,T)}^U(y_t|y_0, y_T) \pi_{0,T}(\mathrm{d}y_0, \mathrm{d}y_T) \ , \tag{15}$$

with $p_{t|(0,T)}^U$ denoting the conditional density of $\mathbf{X}_t$ given $(\mathbf{X}_0, \mathbf{X}_T)$. Also, denote by $\mathrm{K}_t$ the regular kernel associated with the conditional distribution of $(Y_0, Y_T)$ given $Y_t$, *i.e.*, the map $\mathrm{K}_t : \mathbb{R}^d \times \mathcal{B}(\mathbb{R}^{2d}) \to [0, 1]$ such that

  (i)  $y \mapsto \mathrm{K}_t(y, \mathsf{A})$ is measurable for any $\mathsf{A} \in \mathcal{B}(\mathbb{R}^{2d})$;

  (ii)  $\mathsf{A} \mapsto \mathrm{K}_t(y, \mathsf{A})$ is a probability measure for any $y \in \mathbb{R}^d$;

  (iii)  almost surely it holds

$$\mathrm{K}_t(Y_t, \mathsf{A}) := \mathbb{E}\left[\mathbb{1}_\mathsf{A}(Y_0, Y_T)|Y_t\right] \ , \quad \mathsf{A} \in \mathcal{B}(\mathbb{R}^{2d}) \ .$$

It follows again from (5) that $\mathrm{K}_t$ admits a transition density with respect to $\mathrm{Leb}^{2d}$ given by

$$\mathrm{k}_t(y_0, y_T|y_t) = p_{t|(0,T)}^U(y_t|y_0, y_T)(p_t^Y(y_t))^{-1} \pi_{0,T}(y_0, y_T) \ . \tag{16}$$

Furthermore, denote by $\mathbb{P}^{(1)} \in \mathcal{P}(\mathcal{C}_T)$ the law on the Wiener space of the (first update of the) forward Markovian Projection $(X_t^{(1)})_{t \in [0,T]}$ (6) of the stochastic interpolant (5) and by $\tilde{\mathbb{P}}^{(1)} \in \mathcal{P}(\mathcal{C}_T)$ the law on the Wiener space of its time-reversal $(\tilde{X}_t^{(1)})_{t \in [0,T]}$ (21).
We observe that, with the introduced notation, for any $t \in [0, T)$ and $y_t, y_{T-t} \in \mathbb{R}^d$, we have that

$$\begin{aligned}
f_t^{(1)}(y_t) &= 2\int \nabla_{y_t} \log p_{T|t}^U(y_T|y_t) \mathrm{k}_t(\mathrm{d}y_0, \mathrm{d}y_T|y_t) - \nabla U(y_t) \ , \\
g_t^{(1)}(y_{T-t}) &= 2\int \nabla_{y_{T-t}} \log p_{T-t|0}^U(y_{T-t}|y_0) \mathrm{k}_{T-t}(\mathrm{d}y_0, \mathrm{d}y_T|y_{T-t}) + \nabla U(y_{T-t}) \ ,
\end{aligned} \tag{17}$$

where $f^{(1)}$ and $g^{(1)}$ defined in (7) and (13) are drifts of the Markovian projection and its time-reversal.

## B   Technical conditions

We introduce the set of assumptions under which the Markovian projection is well-defined.

**T1.** *For $t > s$, $(s, t, x, y) \mapsto p_{t|s}^U(y|x)$ is continuously differentiable in the $t$ and $s$ variables and twice continuously differentiable in the $x$ and $y$ variables. Furthermore, $\partial_s p_{t|s}^U(y|x)\ \partial_t p_{t|s}^U(y|x)$, $\nabla_x p_{t|s}^U(y|x)$, $\nabla_y p_{t|s}^U(y|x)$, $\nabla_x^2 p_{t|s}^U(y|x)$, $\nabla_y^2 p_{t|s}^U(y|x)$ are bounded.*

Let $\mathbf{b} : \mathbb{R}^d \times (0, T) \to \mathbb{R}^d$ be a given vector field. We make the following assumption:

**T2 (b).** $\mathbf{b} : \mathbb{R}^d \times (0, T) \to \mathbb{R}^d$ *is locally bounded and such that, for at least one probability solution $\mu_t$ to the Fokker-Planck equation*

$$\partial_t \mu_t + \mathrm{div}(\mathbf{b}_t \mu_t) - \Delta \mu_t = 0 \ , \quad t \in (0, T) \ , \quad \mu_0 = \mu \ ,$$

*it holds $\int \|\mathbf{b}_t(x)\|^2 \mu_t(\mathrm{d}x)\mathrm{d}t < +\infty$ .*

**Theorem 3.** *Assume **T**1 and **T**2($f^{(1)}$) with $f^{(1)}$ as in (7). Then, the solution $(X_t^{(1)})_{t\in[0,T]}$ of (6) is such that $X_t^1 \stackrel{dist}{=} Y_t$, for any $t \in [0,T]$.*

**Remark 11.** *We remark that **T**1 and **T**2($f^{(1)}$), with $f^{(1)}$ as in (7) are satisfied in the standard case, i.e., when $U \equiv 0$ and refer to [SCD24, Remark 9] for more details.*

*Proof.* The thesis is a direct consequence of [SCD24, Theorem 1] for $t < T$. The case $t = T$, follows essentially from **T**2($f^{(1)}$), with $f^{(1)}$ as in (7). Indeed, **T**2($f^{(1)}$) implies that for any $\{t_n\}_n$ converging from below to $T$

$$\int_0^T f_t^{(1)}(X_t^{(1)})\mathrm{d}t := \mathrm{L}^2 - \lim_{n\to+\infty} \int_0^{t_n} f_t^{(1)}(X_t^{(1)})\mathrm{d}t ,$$

is well-defined and that any two almost-surely convergent subsequences must have the same limit almost surely (see e.g. [Bal17, Proposition 1.5]). Therefore,

$$\int_0^T f_t^{(1)}(X_t^{(1)})\mathrm{d}t := \mathrm{a.s.} - \lim_{k\to+\infty} \int_0^{t_{n_k}} f_t^{(1)}(X_t^{(1)})\mathrm{d}t ,$$

where $\{t_{n_k}\}_k$ denotes any almost surely convergent subsequence of $\{t_n\}_n$. This means that $(X_t^{(1)})_{t\in[0,T)}$ can be extended to the closed time interval $[0,T]$ by defining

$$X_T^{(1)} := \mathrm{a.s} - \lim_{k\to+\infty} X_{t_{n_k}}^{(1)} = \mathrm{a.s} - \lim_{k\to+\infty} \left\{ \int_0^{t_{n_k}} f_t^{(1)}(X_t^{(1)})\mathrm{d}t + \sqrt{2}B_{t_{n_k}} \right\} , \qquad (18)$$

where $\{t_{n_k}\}_k$ denotes any almost surely convergent subsequence of $\{t_n\}_n$. Fix $F \in \mathcal{C}^{1,2}([0,T]\times\mathbb{R}^d)$ with bounded derivatives and let $(\mathcal{L}_u^{(1)})_{u\in[0,T)}$ denoting the generator of $(X^{(1)})_{u\in[0,T)}$, *i.e.*,

$$\mathcal{L}_u^{(1)}F_u(x) = \left\langle f_u^{(1)}(x), \nabla F_u(x) \right\rangle + \Delta F_u(x) , \quad u \in [0,T).$$

Using **T**2($f^{(1)}$) and dominated convergence theorem, we get that for any $\{t_n\}_n$ converging from below to $T$

$$\int_0^T (\partial_u + \mathcal{L}_u^{(1)})F_u(X_u^{(1)})\mathrm{d}u := \mathrm{L}^2 - \lim_{n\to+\infty} \int_0^{t_n} (\partial_u + \mathcal{L}_u^{(1)})F_u(X_u^{(1)})\mathrm{d}u ,$$

is well-defined and that any two almost-surely convergent subsequences must have the same limit almost surely (see e.g. [Bal17, Proposition 1.5]). This means that

$$\int_0^T (\partial_u + \mathcal{L}_u^{(1)})F_u(X_u^{(1)})\mathrm{d}u := \mathrm{a.s} - \lim_{k\to+\infty} \int_0^{t_{n_k}} (\partial_u + \mathcal{L}_u^{(1)})F_u(X_u^{(1)})\mathrm{d}u , \qquad (19)$$

where $\{t_{n_k}\}_k$ denotes any almost surely convergent subsequence of $\{t_n\}_n$, is well-defined. As a consequence, for any $F \in \mathcal{C}^{1,2}([0,T]\times\mathbb{R}^d)$ with bounded derivatives, we have that

$$\left( F_t(X_t^{(1)}) - F_0(X_0^{(1)}) - \int_0^t (\partial_u + \mathcal{L}_u^{(1)})F_u(X_u^{(1)})\mathrm{d}u \right)_{t\in[0,T]} \qquad (20)$$

is a martingale. Indeed, for $t < T$, (20) follows from [SCD24, Theorem 1]. Whereas, for $t = T$, by continuity of $F$, (18), (19) and Ito formula, given any almost surely convergent subsequence $\{t_{n_k}\}_k$, we have that

$$F_T(X_T^{(1)}) - F_0(X_0^{(1)}) - \int_0^T (\partial_u + \mathcal{L}_u^{(1)})F_u(X_u^{(1)})\mathrm{d}u$$

$$:= \mathrm{a.s} - \lim_{k\to+\infty} \left\{ F_{t_{n_k}}(X_{t_{n_k}}^{(1)}) - F_0(X_0^{(1)}) - \int_0^{t_{n_k}} (\partial_u + \mathcal{L}_u^{(1)})F_u(X_u^{(1)})\mathrm{d}u \right\}$$

$$= \mathrm{a.s} - \lim_{k\to+\infty} \sqrt{2} \int_0^{t_{n_k}} \nabla F_u(X_u^{(1)})\mathrm{d}B_u$$

$$= \sqrt{2} \int_0^T \nabla F_u(X_u^{(1)})\mathrm{d}B_u ,$$

where, in the very last step, we used dominated convergence theorem. Condition (20) makes $(X_t^{(1)})_{t\in[0,T]}$ a diffusion on the closed time interval $[0,T]$. Moreover, we have that $X_T^{(1)} \sim \nu$: by continuity and (9), as $t \to T^-$, we have that $X_t^{(1)} \stackrel{dist}{=} Y_t \Rightarrow Y_T$ and $Y_T \sim \nu$. $\qquad\square$

## C  Geometric Perspective on Iterative Markovian Fitting

For any measure $\mathbb{P} \in \mathcal{P}(\mathcal{C}_T)$, denote by $b\mathbb{P}$ the bridge associated with it, *i.e.*, the Markov kernel on $\mathbb{R}^{2d} \times \mathcal{C}_T$ satisfying for any $A \in \mathcal{B}(\mathcal{C}_T)$, $\mathbb{P}(A) = \int d\mathbb{P}_{0,T}(x_0, x_T) b\mathbb{P}((x_0, x_T), A)$. From a geometric perspective, the Iterative Markovian Fitting (IMF) algorithm alternates between two different projections: constructing the stochastic interpolant corresponds to a projection onto the reciprocal class $\mathcal{R}^U$ of the Langevin diffusion (1) defined as

$$\mathcal{R}^U := \left\{ \mathbb{P} \in \mathcal{P}(\mathcal{C}_T) \ : \ b\mathbb{P} = bR^U \right\} ,$$

while computing the Markovian projection corresponds to a projection onto the space $\mathcal{M}$ of diffusion Markovian processes. To make this structure explicit, we denote, for any given path measure $\mathbb{Q} \in \mathcal{P}(\mathcal{C}_T)$, these projections respectively by $\text{proj}_{\mathcal{R}^U}(\mathbb{Q})$ and $\text{proj}_{\mathcal{M}}(\mathbb{Q})$. They hold the followings:

$$\text{proj}_{\mathcal{R}^U}(\mathbb{Q}) = \arg\min\{\text{KL}(\mathbb{Q}|\mathbb{P}) \ : \ \mathbb{P} \in \mathcal{R}^U\} , \quad \text{proj}_{\mathcal{M}}(\mathbb{Q}) = \arg\min\{\text{KL}(\mathbb{Q}|\mathbb{P}) \ : \ \mathbb{P} \in \mathcal{M}\} .$$

We refer to [SDBCD24, Propositions 2,4] for the detailed proofs. This means that at each iteration, the IMF algorithm moves back and forth between these two constraint sets:

---

**Algorithm 2:** Iterative Markovian Fitting (Geometric Interpretation)

---

**Input:** Initial coupling $\pi_{0,T}^{(0)} \in \Pi(\mu, \nu)$ and Langevin bridge $bR^U$ associated with (1).
**For each iteration** $k = 0, \ldots, N-1$:
**Step 1. Stochastic Interpolant Update**
    Set
$$\Pi^{(k+1)} \leftarrow \text{proj}_{\mathcal{R}^U} \left( \int d\pi_{0,T}^{(k)}(x_0, x_T) bR^U((x_0, x_T), \cdot) \right) .$$

**Step 2. Markovian Projection Update**
    Set
$$\mathbb{P}^{(k+1)} \leftarrow \text{proj}_{\mathcal{M}}(\Pi^{(k+1)}) , \quad \pi_{0,T}^{(k+1)} = \mathbb{P}_{0,T}^{(k+1)} .$$

**End For**
**Output:** The path-distribution $\mathbb{P}^{(N)}$ and the coupling $\pi_{0,T}^{(N)} \in \Pi(\mu, \nu)$.

---

This formulation highlights the alternating nature of the method: each iteration refines the coupling by first enforcing Markovian consistency (via the Markovian projection) and then reintroducing the structure of Langevin bridges (via the stochastic interpolant). This iterative refinement process enables IMF to approximate the SB efficiently. Indeed, the Schrödinger Bridge $\mathbb{P}^\star$ defined in Section 2 is the unique fixed point of the above iterative scheme, *i.e.*, the unique path-measure on $\mathcal{C}_T$ such that

$$\mathbb{P}^\star \in \mathcal{R}^U \cap \mathcal{M} .$$

For a rigorous proof, we refer the reader to [Léo13] and [LRZ14].

### C.1  Comparison with Sinkhorn and Diffusion Schrödinger Bridge

The geometric formulation (2) highlights that IMF naturally integrates principles from the Iterative Proportional Fitting (IPF) method [For40a, Kul68b], also known as Sinkhorn algorithm, a classical technique for approximating Schrödinger bridges. Indeed, both algorithms alternate between projecting onto two different sets of constraints: for IMF, they correspond to Markovian and reciprocal processes, whereas for IPF these sets correspond to measures with prescribed first and second marginals. As a result, one key difference is that while IMF provides, at each iteration, a coupling between the two distributions, Sinkhorn does not. A true coupling is only recovered at convergence. Moreover, when at least one of the distributions is continuous, Sinkhorn algorithm can not be implemented exactly: one must rely on approximations, for instance, parameterizing the potentials via kernel expansions [GCPB16] or neural networks [SDF$^+$17]. This is precisely why the Diffusion Schrödinger Bridge (DSB) [DBTHD21a] algorithm has been adopted. Such an algorithm can be seen as a dynamical formulation of Sinkhorn and requires similar approximations as those used in IMF. However, from a practical perspective, [FVEC22] highlighted the "forgetting" issue of DSB,

where drift errors accumulate over iterations, causing intermediate models to lose alignment with the true Schrödinger bridge. In contrast, as observed by [SDBCD24, Appendix F], IMF mitigates this issue by explicitly projecting onto the reciprocal class of the reference measure, thereby maintaining the correct bridge structure at each iteration and reducing bias accumulation. Moreover, IMF requires only samples at the initial and terminal times, reconstructing intermediate trajectories through the reference bridge, which reduces memory and computational overhead compared to Sinkhorn-based methods that cache all intermediate trajectory samples.

## D   Diffusion Schrödinger Bridge Matching

As seen in Section 2, Diffusion Schrödinger Bridge Matching (DSBM) leverages time-reversals of Markovian projections to mitigate the bias accumulation on the second marginal $\nu$ during practical implementation of Iterative Markovian Fitting (IMF). As shown in [SCD24, Theorem 1] and recalled in Section 2, the mimicking drift $f_t^{(1)}(x)$ can be rewritten as a conditional expectation:

$$f_t^{(1)}(y_t) = \mathbb{E}\left[\overrightarrow{\phi}_t(Y_t, Y_T)\big|Y_t = y_t\right] - \nabla U(y_t) \,, \quad \overrightarrow{\phi}_t(y_t, y_T) = 2\nabla_{y_t} \log p_{T|t}^U(y_T|y_t) \,.$$

As a consequence, it can be approximated via an $L^2$ projection [Kle13, Corollary 8.17], by minimizing:

$$\theta \mapsto \int_0^T \mathbb{E}\left[\left\|f_\theta^{(1)}(t, Y_t) - f_t^{(1)}(Y_t)\right\|^2\right] \mathrm{d}t \,,$$

over a rich enough family of parametrized functions $\{(t, x) \mapsto f_\theta^{(1)}(t, x)\}_{\theta \in \Theta}$. Remarkably, the same holds for the drift of the time-reversal of the Markovian projection $(X_t^{(1)})_{t \in [0,T]}$. Indeed, it holds the following result.

**Proposition 1.** *The time-reversal* $(\tilde{X}_t^{(1)})_{t \in [0,T]}$ *of the Markovian projection* $(X_t^{(1)})_{t \in [0,T]}$ (6) *evolves accordingly to*

$$\mathrm{d}\tilde{X}_t^{(1)} = g_t^{(1)}(\tilde{X}_t^{(1)})\mathrm{d}t + \sqrt{2}\mathrm{d}B_t \,, \quad t \in [0, T) \,, \quad \tilde{X}_0^{(1)} \sim \nu \,, \tag{21}$$

*where* $(B_t)_{t \in [0,T)}$ *is a Brownian motion as in [HP86, Remark 2.5], for any* $t \in [0, T)$, *the drift function* $g_t$ *is given, for any* $y_{T-t} \in \mathbb{R}^d$, *by*

$$g_t^{(1)}(y_{T-t}) = \mathbb{E}\left[\overleftarrow{\phi}_t(Y_0, Y_{T-t})\big|Y_{T-t} = y_{T-t}\right] + \nabla U(y_{T-t}) \,,$$

*and, for any* $t \in [0, T)$, *the vector field* $\overleftarrow{\phi}_t$ *is defined, for any* $y_0, y_{T-t} \in \mathbb{R}^d$, *by*

$$\overleftarrow{\phi}_t(y_0, y_{T-t}) = 2\nabla_{y_{T-t}} \log p_{T-t|0}^U(y_{T-t}|y_0) \,.$$

**Remark 12.** *While the Brownian motion in* (21) *differs from that in* (6), *since our subsequent analysis is conducted purely in terms of distributions, it can be identified for simplicity.*

*Proof of Proposition 1:* It follows from [And82] and Theorem 3 that the time-reversal of $(X_t^{(1)})_{t \in [0,T]}$ (6) evolves accordingly to an SDE with drift function given by

$$-f_{T-t}^{(1)}(y_t) + 2\nabla \log p_{T-t}^Y(y_t) \,,$$

and volatility term equals to $\sqrt{2}$. Therefore, it remains to show that

$$g_{T-t}^{(1)}(y_t) = \mathbb{E}\left[\overleftarrow{\phi}_{T-t}(Y_0, Y_t)\big|Y_t = y_t\right] + \nabla U(y_t) = -f_t^{(1)}(y_t) + 2\nabla \log p_t^Y(y_t) \,.$$

Observe that, as a consequence of (15), we have that

$$\nabla \log p_t^Y(y_t) = \frac{\int_{\mathbb{R}^d \times \mathbb{R}^d} \nabla_{y_t} \left\{\log p_{t|0}^U(y_t|y_0) + \log p_{T|t}^U(y_T|y_t)\right\} \frac{p_{t|0}^U(y_t|y_0) p_{T|t}^U(y_T|y_t)}{p_{T|0}^U(y_T|y_0)} \pi_{0,T}(\mathrm{d}y_0, \mathrm{d}y_T)}{p_t^Y(y_t)} \,.$$

On the other hand, by using (17) and (16), we get that

$$f_t^{(1)}(y_t) = 2\frac{\int_{\mathbb{R}^d \times \mathbb{R}^d} \nabla_{y_t} \log p_{T|t}^U(y_T|y_t) \frac{p_{t|0}^U(y_t|y_0) p_{T|t}^U(y_T|y_T)}{p_{T|0}^U(y_T|y_0)} \pi_{0,T}(\mathrm{d}y_0, \mathrm{d}y_T)}{p_t^Y(y_t)} - \nabla U(y_t) \ .$$

Therefore, we have that

$$\begin{aligned}
&- f_t^{(1)}(y_t) + 2\nabla \log p_t^Y(y_t) \\
&= 2\frac{\int_{\mathbb{R}^d \times \mathbb{R}^d} \nabla_{y_t} \log p_{t|0}^U(y_t|y_0) \frac{p_{t|0}^U(y_t|y_0) p_{T|t}^U(y_T|y_t)}{p_{T|0}^U(y_T|y_0)} \pi_{0,T}(\mathrm{d}y_0, \mathrm{d}y_T)}{p_t^Y(y_t)} + \nabla U(y_t) \ .
\end{aligned}$$

It follows from (16) that

$$- f_t^{(1)}(y) + 2\nabla \log p_t^Y(y_t) = \mathbb{E}\left[\overleftarrow{\phi}_{T-t}(Y_0, Y_t)\big| Y_t = y_t\right] + \nabla U(y_t) \ ,$$

which concludes the proof. $\qquad\qquad\square$

Using again [Kle13, Corollary 8.17], we can therefore approximate also the drift function $g^{(1)}$ in (21) by minimizing a standard L$^2$-functional:

$$\theta \mapsto \int_0^T \mathbb{E}\left[\left\|g_\theta^{(1)}(t, Y_{T-t}) - g_t^{(1)}(Y_{T-t})\right\|^2\right] \ ,$$

over a rich enough family of parametrized functions $\{(t, x) \mapsto g_\theta^{(1)}(t, x)\}_{\theta \in \Theta}$. The resulting DSBM algorithm looks as follows.

# E  Proof of the main results

## E.1  Preliminaries on Functional Inequalities.

In this section, we recall some well-known definitions and results on functional inequalities that will be useful later on.

**Definition 1.** *We say that a measure $\hat{p} \in \mathcal{P}(\mathbb{R}^d)$ satisfies Talagrand inequality with parameter $\xi > 0$, T2($\xi$), if, for any $p \in \mathcal{P}(\mathbb{R}^d)$, it holds*

$$\mathscr{W}_2^2(p, \hat{p}) \leq \xi^{-1}\mathrm{KL}(p|\hat{p}) \ .$$

This inequality was originally introduced by Talagrand [Tal96] for Gaussian measures. Blower in [Blo03] gave a refinement and proved that $\xi$–strong log–concavity, *i.e.*, $\nabla^2(-\log \hat{p}) \geq \xi \mathbb{I}_d$, leads to T2($\xi/2$).

**Definition 2.** *We say that a measure $\hat{p} \in \mathcal{P}(\mathbb{R}^d)$ satisfies log-Sobolev inequality with parameter $\xi > 0$, LSI($\xi$), if, for any $p \in \mathcal{P}(\mathbb{R}^d)$, it holds*

$$\mathrm{KL}(p|\hat{p}) \leq \xi^{-1}\mathscr{I}(p|\hat{p}) \ .$$

It is worth to mention that [CLP23, Theorem 5.7] shows that asymptotic positivity of the integrated convexity profile of the log-density of a probability measure is enough to establish that the measure is a Lipschitz image of a Gaussian and satisfies a LSI. More precisely, it shows that if

$$k_{-\log \mathrm{d}\hat{p}/\mathrm{d}\gamma^d}(r) \geq \hat{\alpha} - r^{-1}\vartheta_{\hat{L}}(r) \ ,$$

for some $\hat{\alpha}, \hat{L} > 0$, then $\hat{p}$ satisfies LSI($\xi$) with

$$\xi = 2(\hat{\alpha} + 1) \Big/ \exp\left(\frac{\hat{L}}{1 + \hat{\alpha}}\right) \ .$$

Moreover, it is well known that if $\hat{p}$ satisfies LSI($\xi$), then it satisfies T2($\xi$). This result was conjectured by Bobkov and Gotze in [BG99] and first proved by Otto and Villani in [OV00].

---

**Algorithm 3:** Diffusion Schrödinger Bridge Matching

---

**Input:** Initial coupling $\pi_{0,T}^{(0)} \in \Pi(\mu, \nu)$ and Langevin bridge $\mathrm{bR}^U$ associated with (1)

**For each iteration** $k = 0, \ldots, N-1$:

  **Step 1. Stochastic Interpolant Update.**

    Define $(Y_t^{(2k+1)})_{t \in [0,T]}$ as:

$$\left(Y_0^{(2k+1)}, Y_T^{(2k+1)}\right) \sim \pi_{0,T}^{(2k)}, \quad Y_{[0,T]}^{(2k+1)} \big| \left(Y_0^{(2k+1)}, Y_T^{(2k+1)}\right) \sim \mathrm{bR}^U\left(\left(Y_0^{(2k+1)}, Y_T^{(2k+1)}\right), \cdot\right).$$

  **Step 2. Forward Markovian Projection Update.**

    • Consider $\mathbb{P}^{(2k+1)}$ as the path-distribution of the solution $(X_t^{(2k+1)})_{t \in [0,T]}$ to the SDE:

$$\mathrm{d}X_t^{(2k+1)} = f_{\theta^\star}^{(2k+1)}(t, X_t^{(2k+1)})\mathrm{d}t + \sqrt{2}\mathrm{d}B_t, \quad X_0^{(2k+1)} \sim \mu,$$

    where $\theta^\star$ is the minimizer of

$$\theta \mapsto \int_0^T \mathbb{E}\left[\left\|f_\theta^{(2k+1)}\left(t, Y_t^{(2k+1)}\right) - f_t^{(2k+1)}\left(Y_t^{(2k+1)}\right)\right\|^2\right],$$

    with

$$f_t^{(2k+1)}(y) = \mathbb{E}\left[\overrightarrow{\phi}_t\left(Y_t^{(2k+1)}, Y_T^{(2k+1)}\right) \big| Y_t^{(2k+1)} = y\right] - \nabla U(y),$$

    and $\overrightarrow{\phi}_t$ is given in (8).

    • Set $\pi_{0,T}^{(2k+1)} = \mathbb{P}_{0,T}^{(2k+1)}$.

  **Step 3. Stochastic Interpolant Update.**

    Define $(Y_t^{(2k+2)})_{t \in [0,T]}$ as:

$$\left(Y_0^{(2k+2)}, Y_T^{(2k+2)}\right) \sim \pi_{0,T}^{(2k+1)}, \quad Y_{[0,T]}^{(2k+2)} \big| \left(Y_0^{(2k+2)}, Y_T^{(2k+2)}\right) \sim \mathrm{bR}^U\left(\left(Y_0^{(2k+2)}, Y_T^{(2k+2)}\right), \cdot\right).$$

  **Step 4. Backward Markovian Projection Update.**

    • Consider $(\mathbb{P}^{(2k+2)})^{\mathrm{R}}$ as the path-distribution of the solution $(X_t^{(2k+2)})_{t \in [0,T]}$ to the SDE:

$$\mathrm{d}X_t^{(2k+2)} = g_{\theta^\star}^{(2k+2)}(t, X_t^{(2k+2)})\mathrm{d}t + \sqrt{2}\mathrm{d}B_t, \quad X_0^{(2k+2)} \sim \nu,$$

    where $\theta^\star$ is the minimizer of

$$\theta \mapsto \int_0^T \mathbb{E}\left[\left\|g_\theta^{(2k+2)}\left(t, Y_{T-t}^{(2k+2)}\right) - g_t^{(2k+2)}\left(Y_{T-t}^{(2k+2)}\right)\right\|^2\right],$$

    with

$$g_t^{(2k+2)}(y) = \mathbb{E}\left[\overleftarrow{\phi}_t\left(Y_0^{(2k+2)}, Y_{T-t}^{(2k+2)}\right) \big| Y_{T-t}^{(2k+2)} = y\right] + \nabla U(y),$$

    and $\overrightarrow{\phi}_t$ is given in (8).

    • Set $\pi_{0,T}^{(2k+2)} = \mathbb{P}_{0,T}^{(2k+2)}$.

**End For**

**Output:** The path-distributions $\mathbb{P}^{(2N-1)}, \mathbb{P}^{(2N)}$ and the couplings $\pi_{0,T}^{(2N-1)}, \pi_{0,T}^{(2N)}$.

---

## E.2 Contractivity Properties of the Markovian Projection

In the sequel, we derive contractivity properties for Markovian Projections. Specifically, we show that if two couplings $\pi_{0,T}, \hat{\pi}_{0,T} \in \Pi(\mu, \nu)$ between $\mu$ and $\nu$ are close in terms of KL-divergence, then their Markovian projections are even closer in KL-divergence.

In this section, for different objects considered in Algorithm 1 starting from $\hat{\pi}_{0,T} \in \Pi(\mu, \nu)$, e.g., the stochastic interpolant, the corresponding Markovian projection, and their associated mimicking drifts, laws, marginals, and kernels, we adopt the same notations as those introduced for $\pi_{0,T}$, with the addition of a hat symbol. Specifically, if an object was denoted by $[\cdot]$ in relation to $\pi_{0,T}$, it will now be denoted by $\hat{[\cdot]}$ for $\hat{\pi}_{0,T}$. In particular, we denote by $\hat{\pi}_{0,T}^{(1)}$, the first iteration of Algorithm 1 starting from $\hat{\pi}_{0,T}$.

**Theorem 4.** *Assume **H1** to **H3**. Let $\pi_{0,T}, \hat{\pi}_{0,T} \in \Pi(\mu, \nu)$. Assume that for any $s \in (0, T)$ and any $y \in \mathbb{R}^d$, the Markov kernel $\hat{K}_s(y, \cdot) \in \mathcal{P}(\mathbb{R}^{2d})$ associated to the stochastic interpolant (5) built on $\hat{\pi}_{0,T}$ and the bridge $\mathrm{bR}^U$ associated to (1) satisfies $\mathrm{T2}(\xi)$, for some $\xi > 0$. Then, it holds*

$$\mathrm{KL}\left(\pi_{0,T}^{(1)} \Big| \hat{\pi}_{0,T}^{(1)}\right) \le \frac{1}{2\xi T} \mathrm{KL}\left(\pi_{0,T} \big| \hat{\pi}_{0,T}\right) .$$

*Proof of Theorem 4:* From the data processing inequality [Nut, Lemma 1.6], we get

$$\mathrm{KL}\left(\pi_{0,T}^{(1)} \Big| \hat{\pi}_{0,T}^{(1)}\right) \le \mathrm{KL}\left(\mathbb{P}_{[0,T]}^{(1)} \Big| \hat{\mathbb{P}}_{[0,T]}^{(1)}\right) .$$

From the standard decomposition of the KL divergence based on Girsanov theorem [CLL23, CCL$^+$22, CDS25, SCD24], we obtain

$$\mathrm{KL}\left(\mathbb{P}_{[0,T]}^{(1)} \Big| \hat{\mathbb{P}}_{[0,T]}^{(1)}\right)$$
$$= \mathrm{KL}(\mu|\mu) + \frac{1}{4}\int_0^{T/2} \mathbb{E}\left[\left\|f_t^{(1)} - \hat{f}_t^{(1)}\right\|^2 (X_t^{(1)})\right] \mathrm{d}t + \frac{1}{4}\int_{T/2}^{T} \mathbb{E}\left[\left\|f_t^{(1)} - \hat{f}_t^{(1)}\right\|^2 (X_t^{(1)})\right] \mathrm{d}t .$$

By the additive property [Léo13, Appendix A] of the KL divergence, we have that

$$\mathrm{KL}\left(\mathbb{P}_{[T/2,T]}^{(1)} \Big| \hat{\mathbb{P}}_{[T/2,T]}^{(1)}\right)$$
$$= \mathbb{E}\left[\mathrm{KL}\left(\mathcal{L}\left(X_{[T/2,T]}^{(1)} \Big| X_{T/2}^{(1)}\right) \Big| \mathcal{L}\left(\hat{X}_{[T/2,T]}^{(1)} \Big| X_{T/2}^{(1)}\right)\right)\right] + \mathrm{KL}\left(\mathbb{P}_{T/2}^{(1)} \Big| \hat{\mathbb{P}}_{T/2}^{(1)}\right)$$
$$= \frac{1}{4}\int_{T/2}^{T} \mathbb{E}\left[\left\|f_t^{(1)} - \hat{f}_t^{(1)}\right\|^2 (X_t^{(1)})\right] \mathrm{d}t + \mathrm{KL}\left(\mathbb{P}_{T/2}^{(1)} \Big| \hat{\mathbb{P}}_{T/2}^{(1)}\right) .$$

The non-negativity of the KL divergence yields

$$\frac{1}{4}\int_{T/2}^{T} \mathbb{E}\left[\left\|f_t^{(1)} - \hat{f}_t^{(1)}\right\|^2 (X_t^{(1)})\right] \mathrm{d}t$$
$$\le \mathrm{KL}\left(\mathbb{P}_{[T/2,T]}^{(1)} \Big| \hat{\mathbb{P}}_{[T/2,T]}^{(1)}\right) = \mathrm{KL}\left(\tilde{\mathbb{P}}_{[0,T/2]}^{(1)} \Big| \hat{\tilde{\mathbb{P}}}_{[0,T/2]}^{(1)}\right)$$
$$= \mathrm{KL}(\nu|\nu) + \frac{1}{4}\int_0^{T/2} \mathbb{E}\left[\left\|\hat{g}_t^{(1)} - g_t^{(1)}\right\|^2 (\tilde{X}_t^{(1)})\right] \mathrm{d}t = \frac{1}{4}\int_0^{T/2} \mathbb{E}\left[\left\|\hat{g}_t^{(1)} - g_t^{(1)}\right\|^2 (\tilde{X}_t^{(1)})\right] \mathrm{d}t .$$

By fixing $0 < \delta < T/2$ and putting these inequalities together, we get using Fatou's lemma,

$$\mathrm{KL}\left(\hat{\mathbb{P}}_{[0,T]}^{(1)} \Big| \mathbb{P}_{[0,T]}^{(1)}\right)$$
$$\le \frac{1}{4}\int_0^{T/2}\left\{\mathbb{E}\left[\left\|f_t^{(1)} - \hat{f}_t^{(1)}\right\|^2 (X_t^{(1)})\right] + \mathbb{E}\left[\left\|g_t^{(1)} - \hat{g}_t^{(1)}\right\|^2 (\tilde{X}_t^{(1)})\right]\right\} \mathrm{d}t$$
$$\le \frac{1}{4}\liminf_{\delta \to 0}\int_\delta^{T/2}\left\{\mathbb{E}\left[\left\|f_t^{(1)} - \hat{f}_t^{(1)}\right\|^2 (X_t^{(1)})\right] + \mathbb{E}\left[\left\|g_t^{(1)} - \hat{g}_t^{(1)}\right\|^2 (\tilde{X}_t^{(1)})\right]\right\} \mathrm{d}t .$$

Fix $t \in [\delta, T/2]$ and $y \in \mathbb{R}^d$. Using the dual representation of the $\mathscr{W}_1$ distance

$$\mathscr{W}_1(p, \hat{p}) = \sup \left\{ \int h(x)(p - \hat{p})(\mathrm{d}x) \mid h \in \mathrm{Lip}_{\leq 1}(\mathbb{R}^d) \right\} \, ,$$

in the expressions (17) we have for the mimicking drifts and **H**3, we get that

$$\left\| f_t^{(1)}(y) - \hat{f}_t^{(1)}(y) \right\|^2 \leq L_U (T - t)^{-2} \mathscr{W}_1^2 \left( \hat{\mathrm{K}}_t(y, \cdot), \mathrm{K}_t(y, \cdot) \right) \, ,$$

and that

$$\left\| g_t^{(1)}(y) - \hat{g}_t^{(1)}(y) \right\|^2 \leq L_U (T - t)^{-2} \mathscr{W}_1^2 \left( \hat{\mathrm{K}}_{T-t}(y, \cdot), \mathrm{K}_{T-t}(y, \cdot) \right) \, .$$

Since, by assumption, $\hat{\mathrm{K}}_t(y, \cdot)$ and $\hat{\mathrm{K}}_{T-t}(y, \cdot)$ satisfy T2($\xi$) when $t \in [\delta, T/2]$, then we have that

$$\left\| f_t^{(1)}(y) - \hat{f}_t^{(1)}(y) \right\|^2 \leq L_U \xi^{-1} (T - t)^{-2} \, \mathrm{KL} \left( \mathrm{K}_t(y, \cdot) \middle| \hat{\mathrm{K}}_t(y, \cdot) \right) \, ,$$

and that

$$\left\| g_t^{(1)}(y) - \hat{g}_t^{(1)}(y) \right\|^2 \leq L_U \xi^{-1} (T - t)^{-2} \, \mathrm{KL} \left( \mathrm{K}_{T-t}(y, \cdot) \middle| \hat{\mathrm{K}}_{T-t}(y, \cdot) \right) \, .$$

By putting all these bounds together and using (9), we get

$$\mathrm{KL} \left( \pi_{0,T}^{(1)} \middle| \hat{\pi}_{0,T}^{(1)} \right) \leq \liminf_{\delta \to 0} \left\{ \frac{L_U}{4\xi} \int_\delta^{T/2} \left[ \frac{1}{(T - t)^2} \left( \int_{\mathbb{R}^d} \mathrm{KL} \left( \mathrm{K}_t(y, \cdot) | \hat{\mathrm{K}}_t(y, \cdot) \right) \mathrm{d}\Pi_t(y) \right. \right. \right.$$

$$\left. \left. \left. + \int_{\mathbb{R}^d} \mathrm{KL} \left( \mathrm{K}_{T-t}(y, \cdot) | \hat{\mathrm{K}}_{T-t}(y, \cdot) \right) \mathrm{d}\Pi_{T-t}(y) \right) \right] \mathrm{d}t \right\} \, .$$

Now, observe that, for any $s \in (0, T)$, we have that

$$\mathrm{KL} \left( \Pi_{[0,T]} \middle| \hat{\Pi}_{[0,T]} \right) \geq \mathrm{KL} \left( \mathcal{L}(Y_0, Y_s, Y_T) \middle| \mathcal{L}(\hat{Y}_0, \hat{Y}_s, \hat{Y}_T) \right)$$

$$= \mathrm{KL} \left( \Pi_s \middle| \hat{\Pi}_s \right) + \int_{\mathbb{R}^d} \mathrm{KL} \left( \mathrm{K}_s(y, \cdot) \middle| \hat{\mathrm{K}}_s(y, \cdot) \right) \mathrm{d}\Pi_s(y)$$

$$\geq \int_{\mathbb{R}^d} \mathrm{KL} \left( \mathrm{K}_s(y, \cdot) \middle| \hat{\mathrm{K}}_s(y, \cdot) \right) \mathrm{d}\Pi_s(y) \, .$$

This fact follows once again from tha data processing inequality [Nut, Lemma 1.6], the additive property [Léo13, Appendix A] and the non-negativity of the KL divergence. It then follows that

$$\mathrm{KL} \left( \pi_{0,T}^{(1)} \middle| \hat{\pi}_{0,T}^{(1)} \right) \leq \liminf_{\delta \to 0} \left\{ \frac{L_U}{2\xi} \mathrm{KL} \left( \Pi_{[0,T]} \middle| \hat{\Pi}_{[0,T]} \right) \int_\delta^{T/2} \frac{1}{(T - t)^2} \mathrm{d}t \right\} \, .$$

From the additive property of the KL divergence [Léo13, Appendix A] it follows that

$$\mathrm{KL} \left( \Pi_{[0,T]} \middle| \hat{\Pi}_{[0,T]} \right)$$

$$= \mathrm{KL} \left( \pi_{0,T} \middle| \hat{\pi}_{0,T} \right) + \int \mathrm{KL} \left( \mathrm{b}\Pi_{[0,T]}((x_0, x_T), \cdot) \middle| \mathrm{b}\hat{\Pi}_{[0,T]}((x_0, x_T), \cdot) \right) \mathrm{d}\pi_{0,T}(x_0, x_T)$$

$$= \mathrm{KL} \left( \pi_{0,T} \middle| \hat{\pi}_{0,T} \right) + \int \mathrm{KL} \left( \mathrm{b}\mathrm{R}^U((x_0, x_T), \cdot) \middle| \mathrm{b}\mathrm{R}^U((x_0, x_T), \cdot) \right) \mathrm{d}\pi_{0,T}(x_0, x_T) \, ,$$

where we used that, for any $\mathsf{A} \in \mathcal{B}(\mathcal{C}_T)$, they hold $\Pi_{[0,T]}(\mathsf{A}) = \int \mathrm{d}\pi_{0,T}(x_0, x_T) \mathrm{b}\Pi_{[0,T]}((x_0, x_T), \mathsf{A})$ and $\hat{\Pi}_{[0,T]}(\mathsf{A}) = \int \mathrm{d}\hat{\pi}_{0,T}(x_0, x_T) \mathrm{b}\hat{\Pi}_{[0,T]}((x_0, x_T), \mathsf{A})$ together with the fact that $\mathrm{b}\Pi_{[0,T]} = \mathrm{b}\hat{\Pi}_{[0,T]} = \mathrm{b}\mathrm{R}^U$. Therefore, we get

$$\mathrm{KL} \left( \pi_{0,T}^{(1)} \middle| \hat{\pi}_{0,T}^{(1)} \right) \leq \liminf_{\delta \to 0} \left\{ \frac{L_U}{2\xi} \mathrm{KL} \left( \pi_{0,T} \middle| \hat{\pi}_{0,T} \right) \int_\delta^{T/2} \frac{1}{(T - t)^2} \mathrm{d}t \right\}$$

$$= \frac{L_U}{2\xi T} \mathrm{KL} \left( \pi_{0,T} \middle| \hat{\pi}_{0,T} \right) \, .$$

The proof is completed. □

### E.3 Proof of the Main Theorems

In this section we establish exponential convergence for the Iterative Markovian Fitting (IMF) algorithm 1. Before proceeding further, let us point out that the results of this section hold true under more general assumptions on the log-densities of the probability distributions $\mu, \nu$ than the ones considered in the main body.

Specifically, we consider, in place of **H**5 (i) and **H**7 (i) respectively, the following generalized conditions.

**H5.** (i)' There exist $\alpha_\mu, \alpha_\nu \in (0, +\infty), \beta_\mu, \beta_\nu \in (0, +\infty]$ such that for any $x \in \mathbb{R}^d$,

$$\alpha_\mu \operatorname{Id} \preceq \nabla^2(-\log \mathfrak{p}_\mu)(x) \preceq \beta_\mu \operatorname{Id}, \quad \alpha_\nu \operatorname{Id} \preceq \nabla^2(-\log \mathfrak{p}_\nu)(x) \preceq \beta_\nu \operatorname{Id}.$$

**H7.** (i)' There exist $\alpha_\mu, \alpha_\nu \in (0, +\infty), \beta_\mu, \beta_\nu \in (0, +\infty], L_\mu, L_\nu, M_\mu, M_\nu > 0$ such that for any $r > 0$

$$k_\mu(r) \geq \alpha_\mu - r^{-1}\vartheta_{L_\mu}(r), \quad -k_\mu(r) \leq \beta_\mu + r^{-1}\vartheta_{M_\mu}(r),$$
$$k_\nu(r) \geq \alpha_\nu - r^{-1}\vartheta_{L_\nu}(r), \quad -k_\nu(r) \leq \beta_\nu + r^{-1}\vartheta_{M_\nu}(r).$$

**Remark 13.** *Note that when $\beta_\mu, \beta_\nu = +\infty$, **H5** (i)' and **H7** (i)' reduce **H5** (i) and **H7** (i) respectively.*

**Theorem 5.** *Assume **H**1 to 4 and **H**5 (i)'-**H**5 (ii). Let $\{\pi_{0,T}^{(n)}\}_{n \geq 1}$ be the sequence defined in Algorithm 1. If $T > \max\{\alpha_\mu^{-1}, \alpha_\nu^{-1}\}$, then, for any $n \in \mathbb{N}$, it holds*

$$\operatorname{KL}(\pi_{0,T}^{(n)}|\pi^\star) \leq \left(\frac{1}{T(\alpha_\varphi + \alpha_\psi + \alpha)}\right)^n \operatorname{KL}(\pi_{0,T}|\pi^\star),$$

*where*

$$\alpha_\varphi = \frac{1}{2}\left(\alpha_\mu + \sqrt{\alpha_\mu^2 + \frac{4\alpha_\mu}{T^2\beta_\nu}}\right) - \frac{1}{T}, \quad \alpha_\psi = \frac{1}{2}\left(\alpha_\nu + \sqrt{\alpha_\nu^2 + \frac{4\alpha_\nu}{T^2\beta_\mu}}\right) - \frac{1}{T}. \quad (22)$$

**Remark 14.** *Note that when $\beta_\mu, \beta_\nu = +\infty$, $\alpha_\varphi, \alpha_\psi$ defined in (22) reduce to $\alpha_\varphi = \alpha_\mu - T^{-1}$ and $\alpha_\psi = \alpha_\nu - T^{-1}$ respectively. Therefore, Theorem 1 is the limit case when $\beta_\mu, \beta_\nu = +\infty$ of Theorem 5.*

**Theorem 6.** *Assume **H**1 to 3, **H**6 and **H**7 (i)'-**H**7 (ii). Let $\{\pi_{0,T}^{(n)}\}_{n \geq 1}$ be the sequence defined in Algorithm 1. If $T > \max\{\alpha_\mu^{-1}, \alpha_\nu^{-1}\}$, then, for any $n \in \mathbb{N}$, it holds*

$$\operatorname{KL}(\pi_{0,T}^{(n)}|\pi^\star) \leq \left(\frac{1}{T\mathrm{C}_{\varphi,\psi,U}}\right)^n \operatorname{KL}(\pi_{0,T}|\pi^\star),$$

*where*

$$\mathrm{C}_{\varphi,\psi,U} = 4(\alpha_\varphi + \alpha_\psi + \alpha)\left/ \exp\left(\frac{9\max\{L_\mu, L_\nu, L\}}{\alpha_\varphi + \alpha_\psi + \alpha}\right)\right., \quad (23)$$

*with $\alpha_\varphi > \alpha_\mu - T^{-1}$ (resp. $\alpha_\psi > \alpha_\nu - T^{-1}$) smallest fixed point of*

$$\alpha = \alpha_\mu - \frac{1}{T} + \frac{G_{\nu,\mu}(\alpha, 2)}{2T^2}, \quad \left(\text{resp. } \alpha = \alpha_\nu - \frac{1}{T} + \frac{G_{\mu,\nu}(\alpha, 2)}{2T^2},\right) \quad (24)$$

*and*

$$G_{\nu,\mu}(\alpha, u) = \inf\{s \geq 0 : F_{\nu,\mu}(\alpha, s) \geq u\}, \quad F_{\nu,\mu}(\alpha, s) = \beta_\nu s + \frac{s}{T(1+T\alpha)} + \frac{\sqrt{s}\vartheta_{L_\mu}(\sqrt{s})}{(1+T\alpha)^2}$$

*(resp.*

$$G_{\mu,\nu}(\alpha, u) = \inf\{s \geq 0 : F_{\mu,\nu}(\alpha, s) \geq u\}, \quad F_{\mu,\nu}(\alpha, s) = \beta_\mu s + \frac{s}{T(1+T\alpha)} + \frac{\sqrt{s}\vartheta_{L_\nu}(\sqrt{s})}{(1+T\alpha)^2}.\right)$$

**Remark 15.** *Note that when $\beta_\mu, \beta_\nu = +\infty$, the fixed points $\alpha_\varphi, \alpha_\psi$ of (24) are exactly $\alpha_\varphi = \alpha_\mu - T^{-1}$ and $\alpha_\psi = \alpha_\nu - T^{-1}$ respectively. Therefore, Theorem 2 is the limit case when $\beta_\mu, \beta_\nu = +\infty$ of Theorem 6.*

*Proof of Theorem 5:* From [Léo13, Theorem 2.12b] and [LRZ14, Theorem 2.14], the Schrödinger Bridge $\pi^\star$ is a fixed point of the IMF algorithm 1. We thus wish to iteratively apply Theorem 4 with $\hat{\pi}_{0,T} = \pi^\star$. To this aim, we simply need to show that, under the assumptions of Theorem 5, the kernel $\hat{K}_t(y, \cdot)$ associated with the stochastic interpolant built on $\hat{\pi}_{0,T} = \pi^\star$ and the bridge $bR^U$ associated to (1) satisfies T2($\xi$), for some $\xi > 0$ and for any fixed $t \in (0, T)$ and $y \in \mathbb{R}^d$. Fix $t \in (0, T)$ and $y \in \mathbb{R}^d$. Note that, as a consequence of (16) and (4), the kernel $\hat{K}_t(x, \cdot)$ admits a density with respect to the Lebesgue measure given by

$$
\begin{aligned}
&\frac{d\hat{K}_t(y, \cdot)}{d\text{Leb}^{2d}}(y_0, y_T) \\
&\propto p^U_{t|(0,T)}(y|y_0, y_T)\pi^\star(y_0, y_T) \\
&= p^U_{t|(0,T)}(y|y_0, y_T)\exp\left(-\varphi(y_0) - \psi(y_T)\right)R^U_{0,T}(y_0, y_T) \\
&= p^U_{0,t,T}(y_0, y, y_T)\exp\left(-\varphi(y_0) - \psi(y_T)\right) \\
&= \exp\left(-\left\{\varphi(y_0) + \psi(y_T) - \log p^U_{0,t,T}(y_0, y, y_T)\right\}\right) .
\end{aligned}
\tag{25}
$$

When $T > \max\{\alpha_\mu^{-1}, \alpha_\nu^{-1}\}$, the byproduct of conditions **H4**, **H5(i)′** and [Gre24, Theorem 6.6.1] leads to

$$
\nabla^2\varphi \succeq \alpha_\varphi \,\text{Id} , \quad \nabla^2\psi \succeq \alpha_\psi \,\text{Id} ,
\tag{26}
$$

with $\alpha_\varphi, \alpha_\psi$ as in (22). It follows from (25), (26) and **H5 (ii)** the 2$\xi$–strong log–concavity of $\hat{K}_t(y, \cdot)$, with $\xi = (\alpha_\varphi + \alpha_\psi + \alpha)/2$. As seen in Section E.1, the 2$\xi$–strong log–concavity of $\hat{K}_t(y, \cdot)$ combined with [Blo03] yield that $\hat{K}_t(y, \cdot)$ satisfies T2($\xi$). The arbitrariness of $t \in (0, T)$ and $y \in \mathbb{R}^d$ together with the iterative application of Theorem 4 for $\hat{\pi}_{0,T} = \pi^\star$ and $\xi = (\alpha_\varphi + \alpha_\psi + \alpha)/2$ allow to conclude.

$\square$

*Proof of Theorem 6:* Once we prove that, under the assumptions of Theorem 6, the kernel $\hat{K}_t(y, \cdot)$ associated with the stochastic interpolant built on the Schrödinger Bridge $\hat{\pi}_{0,T} = \pi^\star$ and the bridge $bR^U$ associated to (1) still satisfies T2($\xi$), for some $\xi > 0$ and for any fixed $t \in (0, T)$ and $y \in \mathbb{R}^d$, we can proceed as in the proof of Theorem 5 and conclude. Fix $t \in (0, T)$ and $y \in \mathbb{R}^d$. Denote by $\bar{h}_{t,y} := -\log(d\hat{K}_t(y, \cdot)/d\gamma^{2d})$. It follows from (25) that

$$
\bar{h}_{t,y}(y_0, y_T) \propto \varphi(y_0) + \psi(y_T) - \log p^U_{0,t,T}(y_0, y, y_T) + \log\gamma^{2d}(y_0, y_T) .
$$

It follows from **H6** and **H7 (i)′** combined with [Con24, Theorem 1.2] and **H7 (ii)** that for any $r > 0$

$$
k_{\bar{h}_{t,y}}(r) \geq \alpha_\varphi - r^{-1}\vartheta_{L_\mu}(r) + \alpha_\psi - r^{-1}\vartheta_{L_\nu}(r) + \alpha - r^{-1}\vartheta_L(r) - 1 .
$$

Now, observe that, for any $r > 0$, the maps $L \mapsto \tanh(r\sqrt{L}/2)$ and $L \mapsto \vartheta_L(r)$ are increasing:

$$
\partial_L\left(\tanh(r\sqrt{L}/2)\right) = \frac{r}{4\sqrt{L}\cosh^2(r\sqrt{L}/2)} > 0 ,
$$

$$
\partial_L\vartheta_L(r) = \frac{1}{\sqrt{L}}\tanh(r\sqrt{L}/2) + \frac{r}{2}\frac{1}{\cosh^2(r\sqrt{L}/2)} > 0 .
$$

Hence,

$$
\begin{aligned}
\vartheta_{L_\mu}(r) + \vartheta_{L_\nu}(r) + \vartheta_L(r) &\leq 3\vartheta_{\max\{L_\mu, L_\nu, L\}}(r) \\
&= 2\sqrt{9\max\{L_\mu, L_\nu, L\}}\tanh\left(r\sqrt{\max\{L_\mu, L_\nu, L\}/2}\right) \\
&\leq 2\sqrt{9\max\{L_\mu, L_\nu, L\}}\tanh\left(r\sqrt{9\max\{L_\mu, L_\nu, L\}/2}\right) \\
&= \vartheta_{9\max\{L_\mu, L_\nu, L\}}(r) .
\end{aligned}
$$

Therefore, we have that

$$
k_{\bar{h}_{t,y}}(r) \geq (\alpha_\varphi + \alpha_\psi + \alpha - 1) - r^{-1}\vartheta_{9\max\{L_\mu, L_\nu, L\}}(r) .
$$

As seen in Section E.1, this combined with [CLP23, Theorem 5.7] yields that $\hat{K}_t(y, \cdot)$ satisfies $\mathrm{LSI}(C_{\varphi,\psi,U}/2)$ with $C_{\varphi,\psi,U}$ as in (23) and, in turn, [OV00] yields that $\hat{K}_t(y, \cdot)$ satisfies $\mathrm{T2}(C_{\varphi,\psi,\mathrm{U}}/2)$.

$\square$

