# OpenReview forum: "Exponential Convergence Guarantees for Iterative Markovian Fitting"
_NeurIPS.cc/2025/Conference — NeurIPS 2025 poster_

### Official Review · Reviewer_gX7U · 2025-06-25

**Clarity:** 3
**Significance:** 2
**Originality:** 3
**Rating:** 4
**Confidence:** 3

**Summary:**

The paper studies non-asymptotic convergence guarantees for IMF procedure. The exponential-speed convergence is established under suitable assumptions, of which the key ones are strong or weak log-concavity of the reference distributions. The manuscript comes with remarks which connect theoretical layouts with practical use-cases.

**Questions:**

1. Assumptions H4, H6 - need to be explicitly stated that $(x\_0, x_T) \mapsto - \log p_{0,t,T}^U(x_0, x_t, x_T)$ is [properties from these assumptions] hold for each $x_t$

2. Theorem 1 and Theorem 2: It seems we need $T > \max{(\alpha_\mu^{-1}, \alpha_\nu^{-1}, \alpha^{-1})}$, e.g., for the contraction in Theorem 1 holds true.
3. A lot of repeated bibtex records:
  - [AVE22] , [AVE23a] ([AVE23a] does not exists, note that [AVE22] is ICLR-23)
  - [AVE23b] - missed author Nicholas M. Boffi
  - [DBTHD21a], [DBTHD21b]
  - [For40a], [For40b]
  - [Kul68a], [Kul68b]
  - [SDBCD23], [SDBCD24] (One from NeuriPS 23, the other from NeurIPS 24 ???)
  - [VTLL21a], [VTLL21b]

   Double check bibtex files if you generate bibtex records with llms.

4. Theorem 4: From the proof it seems we do not need the bridge $\text{bR}^U$ to satisfy $\text{T2}(\xi)$

5. Line 645: “the byproduct of conditions [...] leads to” - I think for clarity it is better to be more specific and detailed here, so does in line 656 in the proof of Theorem 6.

6. Formula after line 656: $k\_{\overline{h}\_{t, y}}$ -> $k_{\nabla \overline{h}_{t, y}}$.

**Ethical Concerns:**

["NO or VERY MINOR ethics concerns only"]

**Final Justification:**

The paper considers an important and interesting area of theory in the context of modern diffusion bridge models, providing novel results. However, the experimental validation of the obtained bounds is lacking. While I generally appreciate the authors' contribution (and therefore vote for acceptance), I am inclined to borderline due to this concern.

**Limitations:**

Discussed above

**Paper Formatting Concerns:**

The used citations style is not typical for NeurIPS paper template

**Quality:**

2

**Strengths And Weaknesses:**

I found the paper to be more-or-less well-written and the considered problem (convergence of IMF) to be interesting. At the same time, from a practical point of view, paper has no practical implications; from a theoretical point of view - it is interesting to study whether the assumptions needed for the convergence hold true could be relaxed - current assumptions (on reference distributions) seems to be rather restrictive. Overall, I have a good impression of the paper.
More specific points:
- The paper has no experiments validating the theoretical conclusions. In particular, it is reasonable to run DSBM on, e.g., gaussian measures/perturbed gaussian measures to see the convergence rates
- There is a preprint studying the convergence of IMF [Kholkin]. Their analysis seems to be rather restrictive and hold only for gaussians, but anyway they are highly related.

[Kholkin] Kholkin et. al., Diffusion & Adversarial Schrödinger Bridges via Iterative Proportional Markovian Fitting, 2024

---

> ### Author Rebuttal · Authors · 2025-07-30
>
> We thank the reviewer for the overall positive impression and for finding the topic of convergence of IMF both interesting and relevant. We appreciate the comments on the theoretical contributions and the effort to relate them to practical use-cases.
>
> *The paper has no experiments validating the theoretical conclusions. In particular, it is reasonable to run DSBM on, e.g., Gaussian measures/perturbed Gaussian measures to see the convergence rates.*
>
> We thank the reviewer for the suggestion. While numerical validation of the bounds could certainly provide additional insights, such an empirical investigation lies outside the scope of this paper and is left for future work.
>
> *There is a preprint studying the convergence of IMF [Kholkin]. Their analysis seems to be rather restrictive and hold only for gaussians, but anyway they are highly related.*
>
> We thank the reviewer for bringing the preprint [6]. While related in spirit, their work analyzes a different algorithm, the Iterative Proportional Markovian Fitting (IPMF), which blends ideas from both the Iterative Markovian Fitting (IMF) and Iterative Proportional Fitting (IPF) algorithms. Their theoretical guarantees are established under highly restrictive conditions, specifically for one-dimensional Gaussian marginals. In contrast, our analysis focuses on the original IMF algorithm and provides non-asymptotic convergence guarantees under significantly more general assumptions. Nonetheless, we agree that their contribution is related, and we will cite it appropriately in the revised version.
>
> *Questions*
>
> We thank the reviewer for the careful and detailed check.
>
> *Clarity of assumptions H4, H6:*
>
> Regarding assumptions H4 and H6, we will clarify the statements and explicitly specify all relevant quantifiers to ensure full precision.
>
> *Time horizon’s condition in Theorem 1 and Theorem 2:*
>
> Concerning the bounds in Theorems 1 and 2, these hold under the explicit assumption $T > \max \{ \alpha_{\mu}^{-1} , \alpha_{\nu}^{-1} \}$, which is stated in the theorems and guarantees that the convexity parameters (whether strong or weak log-convexity) of the Sinkhorn potentials are well-defined and strictly positive [2]. To achieve a contraction rate strictly less than $1$, stronger conditions are indeed necessary. In the strongly convex case, this requires $T > \max\{ \alpha_\mu^{-1}, \alpha_\nu^{-1}, \alpha^{-1}\}$, and we will correct this in Remark 3 following Theorem 1. We thank the reviewer for identifying this typo. In the weakly convex case, as correctly noted in Remark 6, the condition becomes $T > \max\{ \alpha_\mu^{-1}, \alpha_\nu^{-1}, C^{-1}\}$.
>
> *Repeated or incorrect BibTeX entries:*
>
> Regarding the bibliography, we will carefully review and fix all the entries to address the issues pointed out.
>
> *Possible redundant assumption on the bridge in Theorem 4:*
>
> We thank the reviewer for the comment. We believe there may be a misunderstanding. In Theorem 4, the Talagrand-type inequality is required for the Markov kernel associated with the interpolating process, not for the bridge itself. This distinction is essential and follows directly from the structure of the proof. More precisely, after applying the standard KL decomposition [8], one is led to control an $L^2$-norm of the difference between the mimicking drifts of two projections. These mimicking drifts are expressed as expectations with respect to the corresponding Markov kernels (see Equation (17)), not the bridges. This is why the Talagrand inequality is imposed on the Markov kernels. We then use the dual formulation of the 1-Wasserstein distance, followed by a Talagrand-type inequality, to control the $L^2$-distance between the drifts in terms of a KL divergence. This step is crucial to establishing a contraction estimate in KL, and it entirely bypasses the need to assume any transport inequality for the bridge itself.
>
> *Lines 645 and 656:*
>
> Finally, we will revise the indicated formula and provide a clearer, more detailed explanation of line 645 in the updated version.
>
> [2] Conforti, G. (2024). Weak semiconvexity estimates for Schrödinger potentials and logarithmic Sobolev inequality for Schrödinger bridges. Probability Theory and Related Fields, 189(3), 1045-1071.
>
> [6] Kholkin, S., Ksenofontov, G., Li, D., Kornilov, N., Gushchin, N., Suvorikova, A., ... & Korotin, A. (2024). Diffusion & adversarial schr\" odinger bridges via iterative proportional markovian fitting. arXiv preprint arXiv:2410.02601.
>
> [8] Chen, H., Lee, H., & Lu, J. (2023, July). Improved analysis of score-based generative modeling: User-friendly bounds under minimal smoothness assumptions. In International Conference on Machine Learning (pp. 4735-4763). PMLR.

---

> > ### Comment · Reviewer_gX7U · 2025-08-01
> > **Thank you for the answer**
> >
> > I thank the authors for their response and for the given clarifications.
> > Regarding the numerical experiment, I think that, for a theoretical paper of this kind, a numerical experiment that validates the theory would be very appropriate and would considerably benefit the paper.
> > I will keep my score unchanged.

---

> > > ### Comment · Reviewer_NJSy · 2025-08-01
> > > **Regarding experiments**
> > >
> > > IMF is a known algorithm and there are other papers which offer benchmarks of it against Sinkhorn. This paper is not proposing any algorithmic changes, but rather just offers new theoretical analysis. I do not see why experiments would be needed as it is unlikely they would offer any new insights; rather, the paper could just point to another paper which does show experiments.

---

> > > > ### Author Response · Authors · 2025-08-03
> > > >
> > > > We thank both reviewers for engaging the discussion and their feedback. While we appreciate gX7U-Reviewer’s remarks, we share NJSy-Reviewer’s view that numerical experiments are not essential in this context, as our contribution is theoretical in nature and builds on previously established implementations of IMF (e.g., [1], [9]) that already include empirical validation.

---

> ### Comment · Reviewer_gX7U · 2025-08-04
> **Regarding experiments**
>
> The experiments are needed not to show that IMF does work (it follows from previous art), but to numerically validate the obtained exponential convergence rates in the situations where the necessary theoretical conditions (reference distributions are good enough and time horizon is sufficiently long) are satisfied.
>
> I think it is an important component of any theoretical work which study convergence rates of some numerical procedures. The omission of such experimental validation is a drawback

---

### Official Review · Reviewer_THDL · 2025-07-01

**Clarity:** 3
**Significance:** 3
**Originality:** 3
**Rating:** 5
**Confidence:** 3

**Summary:**

The authors analyze the convergence of the Iterative Markovian Fitting (IMF) algorithm for estimating Schrödinger Bridges. In particular, they develop the first non-asymptotic convergence rates for Iterative Markovian Fitting. An analysis is provided for two important regimes:  strongly and weakly log-concave marginal distributions, where the latter case even captures certain non-convex marginals.

**Questions:**

- Can you please elaborate on the dimension-dependence of your analysis (at least for some relevant examples / instances)?
- Can you please elaborate why assumption H2 is reasonable? When is it satisfied?

**Ethical Concerns:**

["NO or VERY MINOR ethics concerns only"]

**Final Justification:**

The authors have satisfactorily addressed my concerns, and I still think the paper makes a valuable contribution.

**Limitations:**

yes -- the discussion could be strengthened though by highlighting where the key challenges in the current analysis lie that would need to be overcome to analyze more practical variants of Iterative Markovian Fitting.

**Paper Formatting Concerns:**

The paper is mostly well written.  I noticed one typo on line 129: “One of the most famous scheme” -> schemes

**Quality:**

3

**Strengths And Weaknesses:**

Strengths:
- The authors provide the first non-asymptotic convergence result for an important algorithm for generative modeling.
- The new contraction results for Markovian projections could be of independent interest.
- Mostly clear structure and exposition, with assumptions are clearly spelled out. Most of them (with one exception, see below) are discussed in detail and settings provided where they hold.

Weaknesses:
- The main weakness and acknowledged limitation is that the analysis applies to an idealized (impractical) version of the Iterative Markovian Fitting algorithm, which assumes exact projections, and ignores effects of discretization. Nevertheless, I think the contribution is still a meaningful first step given the importance of the problem and interest of the machine learning community in the area.

---

> ### Author Rebuttal · Authors · 2025-07-30
>
> We thank the reviewer for their encouraging and constructive remarks. We are pleased that the reviewer highlights the novelty of our non-asymptotic analysis of IMF in both the strongly and weakly log-concave regimes. We are also grateful for the appreciation of the clarity of our assumptions and exposition, and the potential independent interest of the new contraction results.
>
> *Can you please elaborate on the dimension-dependence of your analysis (at least for some relevant examples / instances)?*
>
> We clarify that the convergence bounds we obtain are dimension-free in the same sense as in the literature on Langevin Monte Carlo: the bounds do not depend explicitly on the ambient dimension, but rather on parameters of the marginal distributions, specifically, their (weak or strong) log-convexity constants. While these parameters are independent of the dimension in some specific settings, they may degrade with dimension in more general scenarios.
>
> *Can you please elaborate why assumption H2 is reasonable? When is it satisfied?*
>
> Regarding assumption H2, when the reference potential $U$ in either null or quadratic, a simple sufficient condition for its validity is that the product measure $\mu \otimes \nu$ is absolutely continuous with respect to the reference measure $R^U$ and that the logarithm of its Radon–Nikodym derivative is integrable, i.e., $\log⁡ \frac{d(\mu\otimes\nu)}{d R^U}\in L^1(\mu\otimes\nu)\$. This condition is standard in optimal transport and Schrödinger bridge literature, see e.g. [5]. Moreover, when $U$ is sufficiently smooth, we expect Assumption H2 to hold whenever $\mu$ and $\nu$ are absolutely continuous with respect to the Lebesgue measure.
>
> [5] Nutz, M. (2021). Introduction to entropic optimal transport. Lecture notes, Columbia University.

---

> > ### Comment · Reviewer_THDL · 2025-08-02
> >
> > Thank you for the clarification.  I will increase my score.

---

> > > ### Author Response · Authors · 2025-08-08
> > >
> > > Thank you far taking the time to evaluate our article. We appreciated your comments and insights, which we shall incorporate in the final version.

---

### Official Review · Reviewer_CFWL · 2025-07-03

**Clarity:** 3
**Significance:** 2
**Originality:** 2
**Rating:** 4
**Confidence:** 3

**Summary:**

In this paper, the authors propose a non-ordinary method for proving the exponential convergence rate of the IMF algorithm under specific log-concave assumptions on marginal distributions and reference dynamics. This makes possible to use the duality of the Kantorovich metric and the Talagran inequality to prove contraction of the Markovian projection in IMF by the Kullback-Leibler divergence. The authros derive two convergence rate results, for strongly and weakly log-concave settings. The constant in the resulting inequality requires the time horizon to be large enough.

**Questions:**

- Would the authors consider reformulating the theorem in terms of the reference process's variance rather than time horizon? This alternative presentation might enhance interpretability and better align with established physical interpretations of the convergence behavior.
- What evidence suggests that large time horizons are fundamentally necessary? Could you provide examples where compression fails for smaller horizons to clarify this requirement?
 - Might the authors' methodology be, in principle, extensible to more general classes of marginal distributions? One wonders if there exist implicit conditions under which such generalizations could hold?
 - Could numerical experiments be added at least to: (1) verify convergence behavior at various time horizons, including sorter ones? (2) Test the practical implications of the marginal distribution, including possibly violating log-concavity constraints?

**Ethical Concerns:**

["NO or VERY MINOR ethics concerns only"]

**Final Justification:**

The authors have addressed my concerns partially. *In my opinion*, the lack of numerical experiments that validate the presented convergence rates is a serious shortcoming of the paper. However, this paper still advances an interesting topic and despite having some gray areas in their results, e.g., small values of $T$, I'd rather see this paper accepted than rejected.

**Limitations:**

The authors have addressed the limitations of their approach in the conclusion.

**Paper Formatting Concerns:**

No major formatting issues in this paper.

**Quality:**

2

**Strengths And Weaknesses:**

**Strengths**
The paper is clearly written and all the required notions are properly introduced. The authors prove a rather nontrivial result for an important class of marginals and a reasonable family of reference measures. The proof of the main technical result (Theorem 4) seemed to be quite an interesting technique.

**Weaknesses**
- First, the dependence of the Markov projection's compression constant on the time horizon appears somewhat unclear. The existing analysis might alternatively suggest that the IMF convergence rate depends primarily on noise variance, as compellingly demonstrated in prior work [1] (see Figure 6, "Dependence on the variance of the prior process"). It would be particularly insightful if the authors could reformulate their main theorem to express the compression constant in terms of the reference process's noise variation while keeping the time horizon fixed. Such refinement could provide a more intuitive interpretation of the convergence behavior.
- Second, the requirement for a sufficiently large time horizon needs additional justification. The manuscript would be significantly strengthened by including explicit examples demonstrating the absence of compression properties for shorter time horizons. The absence of such examples might suggest that this condition reflects limitations of the current proof technique rather than fundamental properties of the iterative procedure.
- Third, the assumptions regarding marginal distributions appear somewhat restrictive. By focusing exclusively on unimodal distributions (log-concave and weakly log-concave families), the current theoretical guarantees exclude potentially important cases of multimodal distributions that frequently arise in practical applications. Addressing this limitation through theoretical extension would enhance the paper's applicability.
- Finally, the absence of numerical validation represents a notable gap in the current research. Empirical results would be particularly valuable for addressing: (1) the behavior of convergence rates for different time horizons, including shorter ones, and (2) the depended of convergence for the different marginal distributions,  possibly including the ones that do not fall ini described by authors assumptions. Including such experimental validation would substantially strengthen confidence in the theoretical findings.

These suggested improvements, while not diminishing the paper's theoretical contributions, could help establish a more comprehensive understanding of the method's properties and limitations. The authors' attention to these aspects would likely increase both the depth and practical relevance of their work.


**References.**

[1] N. Gushchin, D. Selikhanovych, S. Kholkin, E. Burnaev, A. Korotin. Adversarial Schrödinger Bridge Matching. The Thirty-eighth Annual Conference on Neural Information Processing Systems. 2024

---

> ### Author Rebuttal · Authors · 2025-07-30
>
> We thank the reviewer for their positive feedback. We appreciate the recognition of the novelty in our approach to analyzing the convergence of IMF under (weak) log-concave assumptions. We are glad the reviewer found our use of Kantorovich duality and Talagrand-type arguments compelling, and that Theorem 4 was seen as a technically interesting contribution.
>
> *On the first weakness: Would the authors consider reformulating the theorem in terms of the reference process's variance rather than time horizon? This alternative presentation might enhance interpretability and better align with established physical interpretations of the convergence behavior.*
>
> We thank the reviewer for this insightful question. There is, in fact, no contradiction between our analysis and the observation in [7] that the convergence behavior depends on the noise variance. This relationship is a direct consequence of the Brownian scaling property: $B_t \sim \sigma B_{t/\sigma^2}$. As a result, our bounds can equivalently be reformulated in terms of the noise variance $\sqrt{2T}$ while fixing the time horizon to $1$, as the reviewer suggests. However, such a reformulation does not alter the asymptotic scaling behavior of the compression constant, which remains of order $O(T^{-1})$. More precisely, if one fixes the time horizon to $1$ and sets the volatility parameter to $\sigma = \sqrt{2T}$, then the mimicking drift appearing in the Markovian projection includes a prefactor of order $1/(1 - t)$. The standard Girsanov decomposition of the KL divergence [8] introduces a multiplicative factor of $\frac{1}{2\sigma^2} = \frac{1}{4T}$, and this ultimately leads to a compression constant of order $O(T^{-1})$, even when the analysis is expressed in terms of the noise variance. We will clarify this equivalence and its implications in the revised manuscript.
>
> *On the second weakness: What evidence suggests that large time horizons are fundamentally necessary? Could you provide examples where compression fails for smaller horizons to clarify this requirement?*
>
> We thank the reviewer for raising this important point. We fully acknowledge that, at present, we do not know whether convergence still holds for small values of $T$. It remains an open and interesting question and we do not currently know if this is a limitation of our current analysis. Nevertheless, we believe our contribution is still significant: to the best of our knowledge, this is the first result establishing geometric convergence for IMF under relatively broad assumptions. While we aimed to provide a comprehensive analysis, scientific progress often advances incrementally. As a point of comparison, geometric convergence for Sinkhorn in compact settings has been known for over a decade, but results in non-compact spaces or with unbounded cost functions have only appeared recently [4].
>
> *On the third weakness: Might the authors' methodology be, in principle, extensible to more general classes of marginal distributions? One wonders if there exist implicit conditions under which such generalizations could hold?*
>
> We thank the reviewer for this important observation. While we acknowledge that extending the theory to any multimodal marginals would be valuable, the assumptions we consider (i.e. log-concavity and weak log-concavity) are, to our knowledge, standard in the literature, particularly when aiming for dimension-free convergence rates. This situation is closely analogous to that of Langevin-based algorithms, where similar assumptions are commonly adopted to obtain geometric convergence bounds that are uniform in dimension. Beyond this setting, convergence guarantees, if they exist, typically depend on the dimension or are no longer geometric. That said, empirically, as also seen with Langevin dynamics, IMF method performs well even outside the log-concave setting (see e.g., [1], [9]).
>
> *On the last weakness: Could numerical experiments be added at least to: (1) verify convergence behavior at various time horizons, including sorter ones? (2) Test the practical implications of the marginal distribution, including possibly violating log-concavity constraints?*
>
> We thank the reviewer for the constructive suggestion. Our work builds upon [1] and [9], who propose practical implementations of IMF, and already include empirical evaluations that address many of the questions raised. Indeed, [1] reports ablation studies on the time horizon, showing improved convergence with larger $T$ in 2D Gaussian settings (Figure 3). Moreover, either [1] and [9] consider non–log-concave marginals, including mixtures, uniform distributions, and real datasets (e.g., MNIST or StyleGAN-generated images), where IMF still performs well despite the lack of theoretical guarantees. That said, in non-Gaussian scenarios, IMF implementations rely on numerical approximations (e.g., of the mimicking drift), which prevent direct verification of convergence rates. In particular, the optimal Schrödinger bridge is not available in closed form. This challenge is not unique to IMF: to the best of our knowledge, there are no empirical studies verifying known convergence rates for Sinkhorn-type methods in continuous, non-Gaussian settings.
>
>
> [1] Shi, Y., De Bortoli, V., Campbell, A., & Doucet, A. (2023). Diffusion schrödinger bridge matching. Advances in Neural Information Processing Systems, 36, 62183-62223.
>
> [4] Chiarini, A., Conforti, G., Greco, G., & Tamanini, L. (2024). A semiconcavity approach to stability of entropic plans and exponential convergence of Sinkhorn's algorithm. arXiv preprint arXiv:2412.09235.
>
> [7] Gushchin, N., Selikhanovych, D., Kholkin, S., Burnaev, E., & Korotin, A. (2024). Adversarial schrödinger bridge matching. Advances in Neural Information Processing Systems, 37, 89612-89651.
>
> [8] Chen, H., Lee, H., & Lu, J. (2023, July). Improved analysis of score-based generative modeling: User-friendly bounds under minimal smoothness assumptions. In International Conference on Machine Learning (pp. 4735-4763). PMLR.
>
> [9] Peluchetti, S. (2023). Diffusion bridge mixture transports, Schrödinger bridge problems and generative modeling. Journal of Machine Learning Research, 24(374), 1-51.

---

> > ### Comment · Reviewer_CFWL · 2025-08-05
> >
> > I thank the authors for taking the time to answer my questions.
> >
> > *On the first weakness*, my concern was fully addressed. Please incorporate an alternative noise variance formulation into the final version of the paper.
> >
> > *On the second weakness*, it is not a good thing that small $T$ convergence results are not available at the moment. In the future would be nice to either prove some results for small values of $T$ or provide an explicit example in which the Markov-projection step fails to contract.
> >
> > *On the third weakness*, log concavity and weak log concavity assumptions restrict the paper's impact. However, I acknowledge that this is the standard practice in the related literature. I would add that it would be beneficial to know where this assumption can be replaced by a milder requirement, such as exponential tail decay of the densities.
> >
> > *On the fourth weakness*, the experiments related to **validation of the convergence rates** are of interest w.r.t. this paper. The [1] and [9] practically validate the IMF convergence, but not the convergence rate. The experiments validating the convergence rate in the setups where all the assumptions are met are strongly desired, e.g., in simple cases where the IMF step can be computed analytically. I agree with reviewer **gX7U** on that part. In contrast, it would be interesting to see the empirical speed of convergence for marginal distributions and time horizons that lie outside your current theoretical framework, but still where IMF step can be computed analytically, if such exist.
> >
> > To conclude, my concerns were partially addressed. Despite some of the mentioned weaknesses that have not been addressed, I'd rather see this paper accepted than rejected. In that light, I raise my score to 4.

---

### Official Review · Reviewer_KTkT · 2025-07-07

**Clarity:** 3
**Significance:** 3
**Originality:** 3
**Rating:** 4
**Confidence:** 3

**Summary:**

The paper analyses the convergence of the (theoretical) IMF scheme to recover the solution of a Schrödinger Bridge (SB) problem for which no quantitative, non-asymptotical results were yet derived. The key ingredient of the proof is to show that the updates are contractive under the assumptions made in the paper. More specifically, in Theorem 4, it is shown that if the Markov kernel associated with the stochastic interpolant of the optimal coupling (which is in fact the solution to the dynamic problem (2) right?) satisfies a Talagrand inequality, then one step of the IMF is contractive. Then, the paper leverages the recent results of Conforti 2024 to recover weak semi-convexity estimates on the optimal coupling. To handle the non-strongly log concave case, the paper uses the fact that weak semi-convexity implies log-Sobolev which in turn implies Talagrand.

**Questions:**

See above + why use IMF rather than Sinkhorn?

**Ethical Concerns:**

["NO or VERY MINOR ethics concerns only"]

**Final Justification:**

I maintain my score. I believe the analysis is most-likely ad-hoc and suboptimal but it has the merit of being the first. Therefore, I am not opposed to acceptance.

**Limitations:**

I feel that the guarantees offered by IMF are overall much weaker than Sinkhorn (especially with $\varepsilon = 1$ as it seems to be the case here) for several reasons:
- Sinkhorn can be implemented \emph{exactly} (I do not agree with l. 129-133) while IMF cannot because the function (11) needs to be approximated and no guarantees are provided with this approximation.
- Sinkhorn converges fast in much larger settings.
In my opinion, this comparison should have been made in the paper.

**Quality:**

3

**Strengths And Weaknesses:**

The strength of the paper is to bring the first quantitative, non-asymptotic guarantees under fairly general assumptions ; remarkably, no uniform log-smoothness assumptions is made.
Weaknesses:
- from a technical point of view, the main contribution seems to be Theorem 4 since the weak semi-convexity estimates were proven in Conforti 2024. Since I am quite far from the SB community, I cannot tell how hard it was to derive (I acknowledge theoretical results always seem obvious once they are derived though).
- because of this proof scheme, that relies on weak-semi-convexity estimates, the quantitative guarantees are in fact quite weak; indeed, weak-log-concavity implies exponentially bad log-Sobolev constants. In particular, how does the exponential bound in Theorem 2 compares to practical settings: does IMF fail when the distributions are not strongly log-concave?

---

> ### Author Rebuttal · Authors · 2025-07-30
>
> We thank the reviewer for their thoughtful summary and detailed appreciation of our contributions. We are particularly grateful for the recognition of our work as the first to provide quantitative, non-asymptotic guarantees for IMF, under general assumptions that avoid uniform log-smoothness.
>
> *From a technical point of view, the main contribution seems to be Theorem 4 since the weak semi-convexity estimates were proven in Conforti 2024. Since I am quite far from the SB community, I cannot tell how hard it was to derive (I acknowledge theoretical results always seem obvious once they are derived though).*
>
> We appreciate the reviewer’s perspective. While the (weak) semi-convexity estimates build upon the prior work [2], our contribution goes significantly beyond this. A central technical novelty of our work lies in deriving a new contraction estimate for the Markovian projection operator (Theorem 4), which is nontrivial and, to the best of our knowledge, has not appeared previously in the literature. Integrating this estimate into a recursive convergence proof under (weak) log-concavity, required careful analysis and constitutes an essential step in establishing non-asymptotic guarantees for IMF.
>
> *Because of this proof scheme, that relies on weak-semi-convexity estimates, the quantitative guarantees are in fact quite weak; indeed, weak-log-concavity implies exponentially bad log-Sobolev constants. In particular, how does the exponential bound in Theorem 2 compares to practical settings: does IMF fail when the distributions are not strongly log-concave?*
>
> We thank the reviewer for their comment. We would like to emphasize that up to our knowledge, estimating a log-Sobolev constant is highly nontrivial in general and only tractable in specific settings. In particular, outside the log-concave case, such constants typically depend exponentially on the parameters of the distribution. Finally, in the literature on Langevin diffusions and gradient-based MCMC methods (see e.g.,[16],[17],[18],[19], and [20]) , both types of assumptions (e.g., convexity vs. functional inequalities) are commonly treated as complementary. Therefore, we do not view our assumptions as a weakness. Furthermore, empirical studies (e.g., [1], [9]) suggest that IMF performs well even when the marginals are not strongly log-concave. This is analogous to MCMC algorithms, where theoretical convergence guarantees can be pessimistic in non-log-concave settings, yet the algorithms remain relatively effective in practice.
>
> *Why use IMF rather than Sinkhorn?*
>
> We thank the reviewer for the question. One key difference is that IMF provides, at each iteration, a coupling between the two distributions, whereas Sinkhorn does not. More specifically, from a primal perspective, Sinkhorn alternates between projections onto sets of joint distributions that match either the first marginal $\mu$ or the second marginal $\nu$, but not both simultaneously. As a result, the intermediate distributions produced by Sinkhorn typically have only one correct marginal at a time. A true coupling is only recovered at convergence. From a practical perspective, [15] highlighted the “forgetting” issue of DSB, where drift errors accumulate over iterations, causing intermediate models to lose alignment with the true Schrödinger bridge. In contrast, as observed by [1] in Appendix F, IMF mitigates this issue by explicitly projecting onto the reciprocal class of the reference measure, thereby maintaining the correct bridge structure at each iteration and reducing bias accumulation. Moreover, IMF requires only samples at the initial and terminal times, reconstructing intermediate trajectories through the reference bridge, which reduces memory and computational overhead compared to Sinkhorn-based methods that cache all intermediate trajectory samples.
>
> *Sinkhorn can be implemented \emph{exactly} (I do not agree with l. 129-133) while IMF cannot because the function (11) needs to be approximated and no guarantees are provided with this approximation.*
>
> We are not aware of any setting in which Sinkhorn can be implemented exactly when at least one of the distributions is continuous. In practice, one must rely on approximations, for instance, parameterizing the potentials via kernel expansions [12] or neural networks [13], which introduces non-trivial computational and analytical challenges (see e.g., [1], [11], [14]). This is precisely why the dynamical formulation of Sinkhorn has been adopted in [1],[11], which also requires similar approximations as those used in IMF. However, to our knowledge, no theoretical results are currently available that account for the approximation errors arising in these practical implementations of the dynamical Sinkhorn approach. In contrast, quantitative convergence results do exist for the ideal (exact) Sinkhorn algorithm. For IMF, such quantitative results were not previously available. Therefore, a necessary first step toward understanding the effect of approximation errors in practical implementations is to establish convergence guarantees at the ideal level. This is exactly what we do in this work.
>
> *Sinkhorn converges fast in much larger settings. In my opinion, this comparison should have been made in the paper.*
>
> We thank the reviewer for this useful remark. We agree that an explicit comparison with Sinkhorn would strengthen the presentation and plan to include it in the revised version of the manuscript. In particular, we will discuss recent results establishing exponential convergence of the Sinkhorn algorithm for weakly log-concave distributions and all values of $T$, with a rate that decays polynomially in $T$ (see [4]).
>
>
> [1] Shi, Y., De Bortoli, V., Campbell, A., & Doucet, A. (2023). Diffusion schrödinger bridge matching. Advances in Neural Information Processing Systems, 36, 62183-62223.
>
> [2] Conforti, G. (2024). Weak semiconvexity estimates for Schrödinger potentials and logarithmic Sobolev inequality for Schrödinger bridges. Probability Theory and Related Fields, 189(3), 1045-1071.
>
> [4] Chiarini, A., Conforti, G., Greco, G., & Tamanini, L. (2024). A semiconcavity approach to stability of entropic plans and exponential convergence of Sinkhorn's algorithm. arXiv preprint arXiv:2412.09235.
>
> [9] Peluchetti, S. (2023). Diffusion bridge mixture transports, Schrödinger bridge problems and generative modeling. Journal of Machine Learning Research, 24(374), 1-51.
>
> [11] Vargas, F., Thodoroff, P., Lamacraft, A., & Lawrence, N. (2021). Solving schrödinger bridges via maximum likelihood. Entropy, 23(9), 1134.
>
> [12] Aude Genevay, Marco Cuturi, Gabriel Peyré, and Francis R. Bach.Stochastic optimization for large-scale optimal transport.In Neural Information Processing Systems, 2016.
>
> [13] Vivien Seguy, Bharath Bhushan Damodaran, Remi Flamary, Nicolas Courty, Antoine Rolet, and Mathieu Blondel.Large-Scale Optimal Transport and Mapping Estimation.In International Conference on Learning Representations, 2018.
>
> [14] Peyré, G., & Cuturi, M. (2019). Computational optimal transport: With applications to data science. Foundations and Trends® in Machine Learning, 11(5-6), 355-607.
>
> [15] Fernandes, D. L., Vargas, F., Ek, C. H., and Campbell, N. D. (2021). Shooting Schrödinger’s cat. In Fourth Symposium on Advances in Approximate Bayesian Inference.
>
> [16] Cheng, X., Chatterji, N. S., Abbasi-Yadkori, Y., Bartlett, P. L., & Jordan, M. I. (2018). Sharp convergence rates for Langevin dynamics in the nonconvex setting. arXiv preprint arXiv:1805.01648.
>
> [17] Bou-Rabee, N., & Eberle, A. (2023). Mixing time guarantees for unadjusted Hamiltonian Monte Carlo. Bernoulli, 29(1), 75-104.
>
> [18] Cheng, X., Zhang, J., & Sra, S. (2022). Efficient sampling on Riemannian manifolds via Langevin MCMC. Advances in Neural Information Processing Systems, 35, 5995-6006.
>
> [19] Ma, Y. A., Chen, Y., Jin, C., Flammarion, N., & Jordan, M. I. (2019). Sampling can be faster than optimization. Proceedings of the National Academy of Sciences, 116(42), 20881-20885.
>
> [20] Chak, M., & Monmarché, P. (2025). Reflection coupling for unadjusted generalized Hamiltonian Monte Carlo in the nonconvex stochastic gradient case. IMA Journal of Numerical Analysis, draf045.

---

> > ### Comment · Reviewer_KTkT · 2025-08-06
> >
> > Thank you for your reply. I will answer point by point:
> > 1) I am still not convinced of the technicality of the proof but ok.
> > 2) I disagree, Langevin does fail on multimodal examples. I think that this is just a technical limitation of your proof scheme.
> > 3/4) In the case where samples are available, I still believe that Sininkhorn, in terms of theoretical guarantees remains a better option. However, it is a good point that in the Bayesian setting, where only un-normalized densities are available, Sinkhorn cannot be computed.
> > 5) Nice.
> > Overall, I believe that the analysis proposed by this paper is probably sub optimal but it has the merit to be the first. I therefore maintain my score.

---

> > > ### Author Response · Authors · 2025-08-08
> > >
> > > Thank you for engaging in the discussion with us. We will incorporate your observations in the final version of the manuscript. In fact, the main goal of the article is not to demonstrate the superiority of IMF against Sinkhorn or vice versa, but rather to start undertaking a theoretical analysis of the convergence properties of IMF.
> > >
> > > Thank you again for the time taken to evaluate our paper. We are happy you could give it a positive evaluation

---

### Official Review · Reviewer_NJSy · 2025-07-22

**Clarity:** 4
**Significance:** 3
**Originality:** 3
**Rating:** 5
**Confidence:** 4

**Summary:**

This work studies the convergence of the IMF method for solving the Schr\”odinger bridge problem. This method consists in alternating Markovian projection steps with stochastic interpolation steps. This method can sometimes outperform Sinkhorn’s algorithm in practice. The paper shows that under log-concavity assumptions on both end marginals as well as the reference path measure, one achieves exponential convergence with rate depending roughly on $1/(\alpha_\mu + \alpha_\nu)$. The result is also extended to the case of ``weak convexity’’.

**Questions:**

**Major**

- I am not particularly familiar with the definition of “weak convexity” used here, so some greater explanation around Theorem 2 would be nice. In particular, is this the expected rate? Can we hope to do better in any meaningful sense?
- Is the large $T$ condition fundamental, or a limitation of the analysis?

**Minor**

L. 177: Steps $\to$ steps

**Ethical Concerns:**

["NO or VERY MINOR ethics concerns only"]

**Final Justification:**

I believe this paper makes a strong contribution as it obtains a relatively sharp (if unsurprising) convergence bound for a canonical algorithm. I believe the lack of experimental validation is NOT a hindrance to this paper's acceptance, and that it should be assessed on the merits of its theoretical contributions only; I am willing to champion its acceptance.

**Limitations:**

Yes.

**Paper Formatting Concerns:**

None.

**Quality:**

3

**Strengths And Weaknesses:**

**Strengths**

- The result (in the strongly convex case) is extremely elegant and stated under interpretable but generic conditions. In particular, the dependence on the convexity parameters $\alpha_\mu, \alpha_\nu$ makes sense.
- As far as I can tell, the proof is also very clean and readable, and may inspire further analysis in this direction. The paper in general is quite well-written.
- I think this is a strong contribution to the literature. As IMF is a well-known algorithm, this will be of interest both to theoreticians and practitioners.

**Weaknesses**

- The algorithms are idealized, and most discretization errors (SDE discretization, estimating the conditional drift) are ignored.
- There is an additional requirement that $T> \alpha^{-1}$ in e.g., Theorem 1, which seems strange. Why would this be necessary?

---

> ### Author Rebuttal · Authors · 2025-07-30
>
> We sincerely thank the reviewer for the very positive evaluation. We are glad that the reviewer finds our convergence result elegant, interpretable, and potentially inspiring for future work. We also appreciate the kind comments on the clarity of the paper and the interest this may generate among both theoreticians and practitioners.
>
> *There is an additional requirement that T>α^{-1}, in e.g., Theorem 1, which seems strange. Why would this be necessary?*
>
> The condition $T>\max\{\alpha_\mu^{-1}, \alpha_\nu^{-1}}$ is a technical assumption required by our proof technique to ensure that the convexity parameters (either strong or weak log-convexity) of the Sinkhorn potentials are well-defined and strictly positive. This is formalized in [2], which we refer to in our analysis.
>
> *I am not particularly familiar with the definition of “weak convexity” used here, so some greater explanation around Theorem 2 would be nice. In particular, is this the expected rate? Can we hope to do better in any meaningful sense?*
>
> Weak log-concavity generalizes log-concavity and includes a broad class of distributions, such as those arising from double-well potentials (Remark 5) and mixture of Gaussian distributions. In fact, weak log-concavity property is also referred to as strong convexity at infinity in the literature related to the convergence of the Langevin diffusion. Indeed, such a hypothesis on the potential of the target distribution has been considered in various works including [10], [16],[17],[18],[19], and [20]. Regarding the rate,  exponential convergence of the Sinkhorn algorithm for weakly log-concave distributions, convergence rate that decays polynomially in $T$, has only been established recently [4]. For IMF, we currently do not know whether our rate or the condition on $T$ can be improved. This remains an open question that we are not able to address.
>
> *Is the large T condition fundamental, or a limitation of the analysis?*
>
> We thank the reviewer for the question. At this stage, we do not know whether the assumption of a large time horizon $T$ is genuinely necessary for convergence, or if it reflects a limitation of our current analysis . In our proof, a large $T$ plays two roles: (i) it guarantees (weak) convexity of the Sinkhorn potentials (as detailed in the previous answers), and (ii) it controls the mimicking drift in the Markovian projection (see the proof of Theorem 4). The sharper condition required for (ii) is discussed in Remarks 3 and 6. Extending the analysis to handle arbitrary or small values of $T$ is an important open problem. We have already spent time investigating this direction, but we currently do not have a viable approach.
>
>
> [2] Conforti, G. (2024). Weak semiconvexity estimates for Schrödinger potentials and logarithmic Sobolev inequality for Schrödinger bridges. Probability Theory and Related Fields, 189(3), 1045-1071.
>
> [4] Chiarini, A., Conforti, G., Greco, G., & Tamanini, L. (2024). A semiconcavity approach to stability of entropic plans and exponential convergence of Sinkhorn's algorithm. arXiv preprint arXiv:2412.09235.
>
> [10] Eberle, A. (2016). Reflection couplings and contraction rates for diffusions. Probability theory and related fields, 166(3), 851-886.
>
> [16] Cheng, X., Chatterji, N. S., Abbasi-Yadkori, Y., Bartlett, P. L., & Jordan, M. I. (2018). Sharp convergence rates for Langevin dynamics in the nonconvex setting. arXiv preprint arXiv:1805.01648.
>
> [17] Bou-Rabee, N., & Eberle, A. (2023). Mixing time guarantees for unadjusted Hamiltonian Monte Carlo. Bernoulli, 29(1), 75-104.
>
> [18] Cheng, X., Zhang, J., & Sra, S. (2022). Efficient sampling on Riemannian manifolds via Langevin MCMC. Advances in Neural Information Processing Systems, 35, 5995-6006.
>
> [19] Ma, Y. A., Chen, Y., Jin, C., Flammarion, N., & Jordan, M. I. (2019). Sampling can be faster than optimization. Proceedings of the National Academy of Sciences, 116(42), 20881-20885.
>
> [20] Chak, M., & Monmarché, P. (2025). Reflection coupling for unadjusted generalized Hamiltonian Monte Carlo in the nonconvex stochastic gradient case. IMA Journal of Numerical Analysis, draf045.

---

> > ### Comment · Reviewer_NJSy · 2025-08-05
> > **Response**
> >
> > I thank the authors for engaging with my comments. I would particularly like to see (perhaps in a follow-up work, as it is out-of-scope of the current work) some more discussion of discretization or approximation errors.
> >
> > It would be helpful if the discussion around weak convexity were added to the text. I am comfortable with this assumption given that a number of previous works have relied on it.
> >
> > My score was already relatively high, and I see no reason to raise it further given the author's feedback. I believe this work has value for theoreticians in this field and I would be happy to push for its acceptance.

---

### Decision · Program_Chairs · 2025-09-17

**Decision:**

Accept (poster)

**Comment:**

The authors proved the convergence of the IMF approach for solving the Schrodinger bridge problem. The paper demonstrates that under the log-concavity assumptions the IMF achieves an exponential convergence rate which depends only on the convexity parameters. Also the authors extended the results to the case of weak convexity.

As a strengths of the paper I would highlight
- conditions of the theorem are sufficiently generic
- the text is well written
- the results is an important step in justification of the IMF algorithm under fairly general assumptions

As a weaknesses I would highlight
- the authors did not take into account discretization or approximation errors
- the dependence of the Markov projection's compression constant on the time horizon and  the requirement for a sufficiently large time horizon
- unimodality of the marginal distributions
- numerical simulations to check the behaviour of convergence rates for different time horizons and marginal distributions
- some additional clarifications should be added to the final text, see comments below

The main reasons to accept are
- the paper provides non-asymptotic guarantees under fairly general assumption for the important algorithm being the first results of this kind. The new contraction results for Markovian projections could be of independent interest
- the paper is mostly well written

During the discussion with the reviewers several issues were raised that should be addressed in the final version of the paper. Here is the list of the main issues
- add discussion around weak convexity (reviewer NJSy)
- discuss benefits of using IMF instead of Sinkhorn (reviewer KTkT)
- discuss reformulation of the theorem in terms of the reference process's variance rather than time horizon; possibility to generalize the results to more general classes of marginal distributions (reviewer CFWL)
- discuss dimension-dependence of the proposed analysis (reviewer THDL)